# Improving MMD-GAN Training with Repulsive Loss Function

**Wei Wang**[*]
University of Melbourne

**Yuan Sun**
RMIT University

**Saman Halgamuge**
University of Melbourne

## Abstract

Generative adversarial nets (GANs) are widely used to learn the data sampling process and their performance may heavily depend on the loss functions, given a limited computational budget. This study revisits MMD-GAN that uses the maximum mean discrepancy (MMD) as the loss function for GAN and makes two contributions. First, we argue that the existing MMD loss function may discourage the learning of fine details in data as it attempts to contract the discriminator outputs of real data. To address this issue, we propose a repulsive loss function to actively learn the difference among the real data by simply rearranging the terms in MMD. Second, inspired by the hinge loss, we propose a bounded Gaussian kernel to stabilize the training of MMD-GAN with the repulsive loss function. The proposed methods are applied to the unsupervised image generation tasks on CIFAR-10, STL-10, CelebA, and LSUN bedroom datasets. Results show that the repulsive loss function significantly improves over the MMD loss at no additional computational cost and outperforms other representative loss functions. The proposed methods achieve an FID score of 16.21 on the CIFAR-10 dataset using a single DCGAN network and spectral normalization. [1]

## 1 Introduction

Generative adversarial nets (GANs) (Goodfellow et al. (2014)) are a branch of generative models that learns to mimic the real data generating process. GANs have been intensively studied in recent years, with a variety of successful applications (Karras et al. (2018); Li et al. (2017b); Lai et al. (2017); Zhu et al. (2017); Ho & Ermon (2016)). The idea of GANs is to jointly train a generator network that attempts to produce artificial samples, and a discriminator network or critic that distinguishes the generated samples from the real ones. Compared to maximum likelihood based methods, GANs tend to produce samples with sharper and more vivid details but require more efforts to train.

Recent studies on improving GAN training have mainly focused on designing loss functions, network architectures and training procedures. The loss function, or simply loss, defines quantitatively the difference of discriminator outputs between real and generated samples. The gradients of loss functions are used to train the generator and discriminator. This study focuses on a loss function called maximum mean discrepancy (MMD), which is well known as the distance metric between two probability distributions and widely applied in kernel two-sample test (Gretton et al. (2012)). Theoretically, MMD reaches its global minimum zero if and only if the two distributions are equal. Thus, MMD has been applied to compare the generated samples to real ones directly (Li et al. (2015); Dziugaite et al. (2015)) and extended as the loss function to the GAN framework recently (Unterthiner et al. (2018); Li et al. (2017a); Bińkowski et al. (2018)).

In this paper, we interpret the optimization of MMD loss by the discriminator as a combination of attraction and repulsion processes, similar to that of linear discriminant analysis. We argue that the existing MMD loss may discourage the learning of fine details in data, as the discriminator attempts to minimize the within-group variance of its outputs for the real data. To address this issue, we propose a repulsive loss for the discriminator that explicitly explores the differences among real data. The proposed loss achieved significant improvements over the MMD loss on image generation

---

[*]Corresponding author: weiw8@student.unimelb.edu.au

[1]The code is available at: `https://github.com/richardwth/MMD-GAN`

tasks of four benchmark datasets, without incurring any additional computational cost. Furthermore, a bounded Gaussian kernel is proposed to stabilize the training of discriminator. As such, using a single kernel in MMD-GAN is sufficient, in contrast to a linear combination of kernels used in Li et al. (2017a) and Bińkowski et al. (2018). By using a single kernel, the computational cost of the MMD loss can potentially be reduced in a variety of applications.

The paper is organized as follows. Section 2 reviews the GANs trained using the MMD loss (MMD-GAN). We propose the repulsive loss for discriminator in Section 3, introduce two practical techniques to stabilize the training process in Section 4, and present the results of extensive experiments in Section 5. In the last section, we discuss the connections between our model and existing work.

## 2 MMD-GAN

In this section, we introduce the GAN model and MMD loss. Consider a random variable $\mathbf{X} \in \mathcal{X}$ with an empirical data distribution $P_{\mathbf{X}}$ to be learned. A typical GAN model consists of two neural networks: a generator $G$ and a discriminator $D$. The generator $G$ maps a latent code $\boldsymbol{z}$ with a fixed distribution $P_{\mathbf{Z}}$ (e.g., Gaussian) to the data space $\mathcal{X}$: $\boldsymbol{y} = G(\boldsymbol{z}) \in \mathcal{X}$, where $\boldsymbol{y}$ represents the generated samples with distribution $P_G$. The discriminator $D$ evaluates the scores $D(\boldsymbol{a}) \in \mathbb{R}^d$ of a real or generated sample $\boldsymbol{a}$. This study focuses on image generation tasks using convolutional neural networks (CNN) for both $G$ and $D$.

Several loss functions have been proposed to quantify the difference of the scores between real and generated samples: $\{D(\boldsymbol{x})\}$ and $\{D(\boldsymbol{y})\}$, including the minimax loss and non-saturating loss (Goodfellow et al. (2014)), hinge loss (Tran et al. (2017)), Wasserstein loss (Arjovsky et al. (2017); Gulrajani et al. (2017)) and maximum mean discrepancy (MMD) (Li et al. (2017a); Bińkowski et al. (2018)) (see Appendix B.1 for more details). Among them, MMD uses kernel embedding $\phi(\boldsymbol{a}) = k(\cdot, \boldsymbol{a})$ associated with a characteristic kernel $k$ such that $\phi$ is infinite-dimensional and $\langle \phi(\boldsymbol{a}), \phi(\boldsymbol{b}) \rangle_{\mathcal{H}} = k(\boldsymbol{a}, \boldsymbol{b})$. The squared MMD distance between two distributions $P$ and $Q$ is

$$M_k^2(P,Q) = \|\boldsymbol{\mu}_P - \boldsymbol{\mu}_Q\|_{\mathcal{H}}^2 = \mathbb{E}_{\boldsymbol{a}, \boldsymbol{a}' \sim P}[k(\boldsymbol{a}, \boldsymbol{a}')] + \mathbb{E}_{\boldsymbol{b}, \boldsymbol{b}' \sim Q}[k(\boldsymbol{b}, \boldsymbol{b}')] - 2\mathbb{E}_{\boldsymbol{a} \sim P, \boldsymbol{b} \sim Q}[k(\boldsymbol{a}, \boldsymbol{b})] \quad (1)$$

The kernel $k(\boldsymbol{a}, \boldsymbol{b})$ measures the similarity between two samples $\boldsymbol{a}$ and $\boldsymbol{b}$. Gretton et al. (2012) proved that, using a characteristic kernel $k$, $M_k^2(P, Q) \geq 0$ with equality applies if and only if $P = Q$.

In MMD-GAN, the discriminator $D$ can be interpreted as forming a new kernel with $k$: $k \circ D(\boldsymbol{a}, \boldsymbol{b}) = k(D(\boldsymbol{a}), D(\boldsymbol{b})) = k_D(\boldsymbol{a}, \boldsymbol{b})$. If $D$ is injective, $k \circ D$ is characteristic and $M_{k \circ D}^2(P_{\mathbf{X}}, P_G)$ reaches its minimum if and only if $P_{\mathbf{X}} = P_G$ (Li et al. (2017a)). Thus, the objective functions for $G$ and $D$ could be (Li et al. (2017a); Bińkowski et al. (2018)):

$$\min_G L_G^{\mathrm{mmd}} = M_{k \circ D}^2(P_{\mathbf{X}}, P_G) = \mathbb{E}_{P_G}[k_D(\boldsymbol{y}, \boldsymbol{y}')] - 2\mathbb{E}_{P_{\mathbf{X}}, P_G}[k_D(\boldsymbol{x}, \boldsymbol{y})] + \mathbb{E}_{P_{\mathbf{X}}}[k_D(\boldsymbol{x}, \boldsymbol{x}')] \quad (2)$$

$$\min_D L_D^{\mathrm{att}} = -M_{k \circ D}^2(P_{\mathbf{X}}, P_G) = 2\mathbb{E}_{P_{\mathbf{X}}, P_G}[k_D(\boldsymbol{x}, \boldsymbol{y})] - \mathbb{E}_{P_{\mathbf{X}}}[k_D(\boldsymbol{x}, \boldsymbol{x}')] - \mathbb{E}_{P_G}[k_D(\boldsymbol{y}, \boldsymbol{y}')] \quad (3)$$

MMD-GAN has been shown to be more effective than the model that directly uses MMD as the loss function for the generator $G$ (Li et al. (2017a)).

Liu et al. (2017) showed that MMD and Wasserstein metric are weaker objective functions for GAN than the Jensen–Shannon (JS) divergence (related to minimax loss) and total variation (TV) distance (related to hinge loss). The reason is that convergence of $P_G$ to $P_{\mathbf{X}}$ in JS-divergence and TV distance also implies convergence in MMD and Wasserstein metric. Weak metrics are desirable as they provide more information on adjusting the model to fit the data distribution (Liu et al. (2017)). Nagarajan & Kolter (2017) proved that the GAN trained using the minimax loss and gradient updates on model parameters is locally exponentially stable near equilibrium, while the GAN using Wasserstein loss is not. In Appendix A, we demonstrate that the MMD-GAN trained by gradient descent is locally exponentially stable near equilibrium.

## 3 REPULSIVE LOSS FUNCTION

In this section, we interpret the training of MMD-GAN (using $L_D^{\mathrm{att}}$ and $L_G^{\mathrm{mmd}}$) as a combination of attraction and repulsion processes, and propose a novel repulsive loss function for the discriminator by rearranging the components in $L_D^{\mathrm{att}}$.

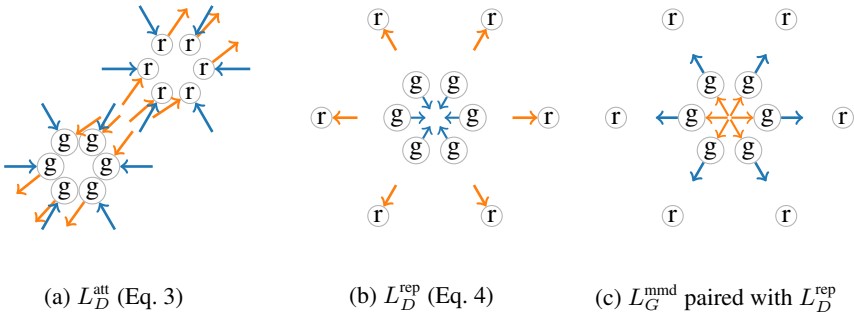

(a) $L_D^{\text{att}}$ (Eq. 3)  (b) $L_D^{\text{rep}}$ (Eq. 4)  (c) $L_G^{\text{mmd}}$ paired with $L_D^{\text{rep}}$

Figure 1: Illustration of the gradient directions of each loss on the real sample scores $\{D(\boldsymbol{x})\}$ ("r" nodes) and generated sample scores $\{D(\boldsymbol{y})\}$ ("g" nodes). The blue arrows stand for attraction and the orange arrows for repulsion. When $L_G^{\text{mmd}}$ is paired with $L_D^{\text{att}}$, the gradient directions of $L_G^{\text{mmd}}$ on $\{D(\boldsymbol{y})\}$ can be obtained by reversing the arrows in (a), thus are omitted.

First, consider a linear discriminant analysis (LDA) model as the discriminator. The task is to find a projection $\boldsymbol{w}$ to maximize the between-group variance $\left\|\boldsymbol{w}^T\boldsymbol{\mu}_x - \boldsymbol{w}^T\boldsymbol{\mu}_y\right\|$ and minimize the within-group variance $\boldsymbol{w}^T(\boldsymbol{\Sigma}_x + \boldsymbol{\Sigma}_y)\boldsymbol{w}$, where $\boldsymbol{\mu}$ and $\boldsymbol{\Sigma}$ are group mean and covariance.

In MMD-GAN, the neural-network discriminator works in a similar way as LDA. By minimizing $L_D^{\text{att}}$, the discriminator $D$ tackles two tasks: 1) $D$ reduces $\mathbb{E}_{P_\mathbf{X}, P_G}[k_D(\boldsymbol{x}, \boldsymbol{y})]$, i.e., causes the two groups $\{D(\boldsymbol{x})\}$ and $\{D(\boldsymbol{y})\}$ to repel each other (see Fig. 1a orange arrows), or maximize between–group variance; and 2) $D$ increases $\mathbb{E}_{P_\mathbf{X}}[k_D(\boldsymbol{x}, \boldsymbol{x}')]$ and $\mathbb{E}_{P_G}[k(\boldsymbol{y}, \boldsymbol{y}')]$, i.e. contracts $\{D(\boldsymbol{x})\}$ and $\{D(\boldsymbol{y})\}$ within each group (see Fig. 1a blue arrows), or minimize the within-group variance. We refer to loss functions that contract real data scores as attractive losses.

We argue that the attractive loss $L_D^{\text{att}}$ (Eq. 3) has two issues that may slow down the GAN training:

1. The discriminator $D$ may focus more on the similarities among real samples (in order to contract $\{D(\boldsymbol{x})\}$) than the fine details that separate them. Initially, $G$ produces low-quality samples and it may be adequate for $D$ to learn the common features of $\{\boldsymbol{x}\}$ in order to distinguish between $\{\boldsymbol{x}\}$ and $\{\boldsymbol{y}\}$. Only when $\{D(\boldsymbol{y})\}$ is sufficiently close to $\{D(\boldsymbol{x})\}$ will $D$ learn the fine details of $\{\boldsymbol{x}\}$ to be able to separate $\{D(\boldsymbol{x})\}$ from $\{D(\boldsymbol{y})\}$. Consequently, $D$ may leave out some fine details in real samples, thus $G$ has no access to them during training.

2. As shown in Fig. 1a, the gradients on $D(\boldsymbol{y})$ from the attraction (blue arrows) and repulsion (orange arrows) terms in $L_D^{\text{att}}$ (and thus $L_G^{\text{mmd}}$) may have opposite directions during training. Their summation may be small in magnitude even when $D(\boldsymbol{y})$ is far away from $D(\boldsymbol{x})$, which may cause $G$ to stagnate locally.

Therefore, we propose a repulsive loss for $D$ to encourage repulsion of the real data scores $\{D(\boldsymbol{x})\}$:

$$L_D^{\text{rep}} = \mathbb{E}_{P_\mathbf{X}}[k_D(\boldsymbol{x}, \boldsymbol{x}')] - \mathbb{E}_{P_G}[k_D(\boldsymbol{y}, \boldsymbol{y}')] \tag{4}$$

The generator $G$ uses the same MMD loss $L_G^{\text{mmd}}$ as before (see Eq. 2). Thus, the adversary lies in the fact that $D$ contracts $\{D(\boldsymbol{y})\}$ via maximizing $\mathbb{E}_{P_G}[k_D(\boldsymbol{y}, \boldsymbol{y}')]$ (see Fig. 1b) while $G$ expands $\{D(\boldsymbol{y})\}$ (see Fig. 1c). Additionally, $D$ also learns to separate the real data by minimizing $\mathbb{E}_{P_\mathbf{X}}[k_D(\boldsymbol{x}, \boldsymbol{x}')]$, which actively explores the fine details in real samples and may result in more meaningful gradients for $G$. Note that in Eq. 4, $D$ does not explicitly push the average score of $\{D(\boldsymbol{y})\}$ away from that of $\{D(\boldsymbol{x})\}$ because it may have no effect on the pair-wise sample distances. But $G$ aims to match the average scores of both groups. Thus, we believe, compared to the model using $L_G^{\text{mmd}}$ and $L_D^{\text{att}}$, our model of $L_G^{\text{mmd}}$ and $L_D^{\text{rep}}$ is less likely to yield opposite gradients when $\{D(\boldsymbol{y})\}$ and $\{D(\boldsymbol{x})\}$ are distinct (see Fig. 1c). In Appendix A, we demonstrate that GAN trained using gradient descent and the repulsive MMD loss ($L_D^{\text{rep}}$, $L_G^{\text{mmd}}$) is locally exponentially stable near equilibrium.

At last, we identify a general form of loss function for the discriminator $D$:

$$L_{D,\lambda} = \lambda\mathbb{E}_{P_\mathbf{X}}[k_D(\boldsymbol{x}, \boldsymbol{x}')] - (\lambda - 1)\mathbb{E}_{P_\mathbf{X}, P_G}[k_D(\boldsymbol{x}, \boldsymbol{y})] - \mathbb{E}_{P_G}[k_D(\boldsymbol{y}, \boldsymbol{y}')] \tag{5}$$

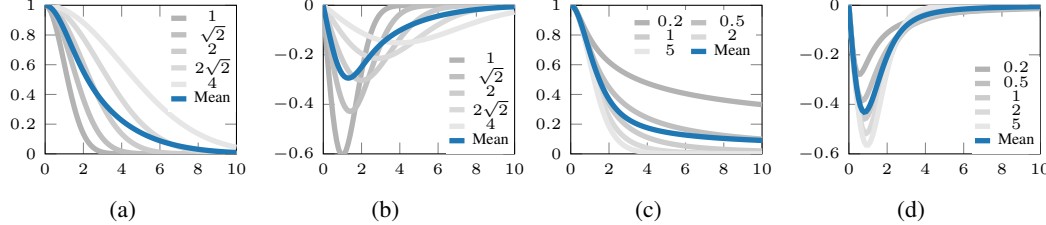

Figure 2: (a) Gaussian kernels $\{k_{\sigma_i}^{\text{rbf}}(\boldsymbol{a}, \boldsymbol{b})\}$ and their mean as a function of $e = \|\boldsymbol{a} - \boldsymbol{b}\|$, where $\sigma_i \in \{1, \sqrt{2}, 2, 2\sqrt{2}, 4\}$ were used in our experiments; (b) derivatives of $\{k_{\sigma_i}^{\text{rbf}}(\boldsymbol{a}, \boldsymbol{b})\}$ in (a); (c) rational quadratic kernel $\{k_{\alpha_i}^{\text{rq}}(\boldsymbol{a}, \boldsymbol{b})\}$ and their mean, where $\alpha_i \in \{0.2, 0.5, 1, 2, 5\}$; (d) derivatives of $\{k_{\alpha_i}^{\text{rq}}(\boldsymbol{a}, \boldsymbol{b})\}$ in (c).

where $\lambda$ is a hyper-parameter[2]. When $\lambda < 0$, the discriminator loss $L_{D,\lambda}$ is attractive, with $\lambda = -1$ corresponding to the original MMD loss $L_D^{\text{att}}$ in Eq. 3; when $\lambda > 0$, $L_{D,\lambda}$ is repulsive and $\lambda = 1$ corresponds to $L_D^{\text{rep}}$ in Eq. 4. It is interesting that when $\lambda > 1$, the discriminator explicitly contracts $\{D(\boldsymbol{x})\}$ and $\{D(\boldsymbol{y})\}$ via maximizing $\mathbb{E}_{P_{\mathbf{X}}, P_G}[k_D(\boldsymbol{x}, \boldsymbol{y})]$, which may work as a penalty that prevents the pairwise distances of $\{D(\boldsymbol{x})\}$ from increasing too fast. Note that $L_{D,\lambda}$ has the same computational cost as $L_D^{\text{att}}$ (Eq. 3) as we only rearranged the terms in $L_D^{\text{att}}$.

## 4 REGULARIZATION ON MMD AND DISCRIMINATOR

In this section, we propose two approaches to stabilize the training of MMD-GAN: 1) a bounded kernel to avoid the saturation issue caused by an over-confident discriminator; and 2) a generalized power iteration method to estimate the spectral norm of a convolutional kernel, which was used in spectral normalization on the discriminator in all experiments in this study unless specified otherwise.

### 4.1 KERNEL IN MMD

For MMD-GAN, the following two kernels have been used:

- Gaussian radial basis function (RBF), or Gaussian kernel (Li et al. (2017a)), $k_{\sigma}^{\text{rbf}}(\boldsymbol{a}, \boldsymbol{b}) = \exp(-\frac{1}{2\sigma^2} \|\boldsymbol{a} - \boldsymbol{b}\|^2)$ where $\sigma > 0$ is the kernel scale or bandwidth.

- Rational quadratic kernel (Bińkowski et al. (2018)), $k_{\alpha}^{\text{rq}}(\boldsymbol{a}, \boldsymbol{b}) = (1 + \frac{1}{2\alpha} \|\boldsymbol{a} - \boldsymbol{b}\|^2)^{-\alpha}$, where the kernel scale $\alpha > 0$ corresponds to a mixture of Gaussian kernels with a Gamma$(\alpha, 1)$ prior on the inverse kernel scales $\sigma^{-1}$.

It is interesting that both studies used a linear combination of kernels with five different kernel scales, i.e., $k_{\text{rbf}} = \sum_{i=1}^{5} k_{\sigma_i}^{\text{rbf}}$ and $k_{\text{rq}} = \sum_{i=1}^{5} k_{\alpha_i}^{\text{rq}}$, where $\sigma_i \in \{1, 2, 4, 8, 16\}$, $\alpha_i \in \{0.2, 0.5, 1, 2, 5\}$ (see Fig. 2a and 2c for illustration). We suspect the reason is that a single kernel $k(\boldsymbol{a}, \boldsymbol{b})$ is saturated when the distance $\|\boldsymbol{a} - \boldsymbol{b}\|$ is either too large or too small compared to the kernel scale (see Fig. 2b and 2d), which may cause diminishing gradients during training. Both Li et al. (2017a) and Bińkowski et al. (2018) applied penalties on the discriminator parameters but not to the MMD loss itself. Thus the saturation issue may still exist. Using a linear combination of kernels with different kernel scales may alleviate this issue but not eradicate it.

Inspired by the hinge loss (see Appendix B.1), we propose a bounded RBF (RBF-B) kernel for the discriminator. The idea is to prevent $D$ from pushing $\{D(\boldsymbol{x})\}$ too far away from $\{D(\boldsymbol{y})\}$ and causing saturation. For $L_D^{\text{att}}$ in Eq. 3, the RBF-B kernel is:

$$k_{\sigma}^{\text{rbf-b}}(\boldsymbol{a}, \boldsymbol{b}) = \begin{cases} \exp(-\frac{1}{2\sigma^2} \max(\|\boldsymbol{a} - \boldsymbol{b}\|^2, b_l)) & \text{if } \boldsymbol{a}, \boldsymbol{b} \in \{D(\boldsymbol{x})\} \text{ or } \boldsymbol{a}, \boldsymbol{b} \in \{D(\boldsymbol{y})\} \\ \exp(-\frac{1}{2\sigma^2} \min(\|\boldsymbol{a} - \boldsymbol{b}\|^2, b_u)) & \text{if } \boldsymbol{a} \in \{D(\boldsymbol{x})\} \text{ and } \boldsymbol{b} \in \{D(\boldsymbol{y})\} \end{cases} \quad (6)$$

---

[2] The weights for the three terms in $L_{D,\lambda}$ sum up to zero. This is to make sure the $\partial L_{D,\lambda} / \partial \boldsymbol{\theta}_D$ is zero at equilibrium $P_{\mathbf{X}} = P_G$, where $\boldsymbol{\theta}_D$ is the parameters of $D$.

For $L_D^{\text{rep}}$ in Eq. 4, the RBF-B kernel is:

$$k_\sigma^{\text{rbf-b}}(\boldsymbol{a}, \boldsymbol{b}) = \begin{cases} \exp(-\frac{1}{2\sigma^2}\max(\|\boldsymbol{a}-\boldsymbol{b}\|^2, b_l)) & \text{if } \boldsymbol{a}, \boldsymbol{b} \in \{D(\boldsymbol{y})\} \\ \exp(-\frac{1}{2\sigma^2}\min(\|\boldsymbol{a}-\boldsymbol{b}\|^2, b_u)) & \text{if } \boldsymbol{a}, \boldsymbol{b} \in \{D(\boldsymbol{x})\} \end{cases} \tag{7}$$

where $b_l$ and $b_u$ are the lower and upper bounds. As such, a single kernel is sufficient and we set $\sigma = 1$, $b_l = 0.25$ and $b_u = 4$ in all experiments for simplicity and leave their tuning for future work. It should be noted that, like the case of hinge loss, the RBF-B kernel is used only for the discriminator to prevent it from being over-confident. The generator is always trained using the original RBF kernel, thus we retain the interpretation of MMD loss $L_G^{\text{mmd}}$ as a metric.

RBF-B kernel is among many methods to address the saturation issue and stabilize MMD-GAN training. We found random sampling kernel scale, instance noise (Sønderby et al. (2017)) and label smoothing (Szegedy et al. (2016); Salimans et al. (2016)) may also improve the model performance and stability. However, the computational cost of RBF-B kernel is relatively low.

## 4.2 Spectral Normalization in Discriminator

Without any Lipschitz constraints, the discriminator $D$ may simply increase the magnitude of its outputs to minimize the discriminator loss, causing unstable training[3]. Spectral normalization divides the weight matrix of each layer by its spectral norm, which imposes an upper bound on the magnitudes of outputs and gradients at each layer of $D$ (Miyato et al. (2018)). However, to estimate the spectral norm of a convolution kernel, Miyato et al. (2018) reshaped the kernel into a matrix. We propose a generalized power iteration method to directly estimate the spectral norm of a convolution kernel (see Appendix C for details) and applied spectral normalization to the discriminator in all experiments. In Appendix D.1, we explore using gradient penalty to impose the Lipschitz constraint (Gulrajani et al. (2017); Bińkowski et al. (2018); Arbel et al. (2018)) for the proposed repulsive loss.

## 5 Experiments

In this section, we empirically evaluate the proposed 1) repulsive loss $L_D^{\text{rep}}$ (Eq. 4) on unsupervised training of GAN for image generation tasks; and 2) RBF-B kernel to stabilize MMD-GAN training. The generalized power iteration method is evaluated in Appendix C.3. To show the efficacy of $L_D^{\text{rep}}$, we compared the loss functions $(L_D^{\text{rep}}, L_G^{\text{mmd}})$ using Gaussian kernel (MMD-rep) with $(L_D^{\text{att}}, L_G^{\text{mmd}})$ using Gaussian kernel (MMD-rbf) (Li et al. (2017a)) and rational quadratic kernel (MMD-rq) (Bińkowski et al. (2018)), as well as non-saturating loss (Goodfellow et al. (2014)) and hinge loss (Tran et al. (2017)). To show the efficacy of RBF-B kernel, we applied it to both $L_D^{\text{att}}$ and $L_D^{\text{rep}}$, resulting in two methods MMD-rbf-b and MMD-rep-b. The Wasserstein loss was excluded for comparison because it cannot be directly used with spectral normalization ( Miyato et al. (2018)).

### 5.1 Experiment Setup

**Dataset:** The loss functions were evaluated on four datasets: 1) CIFAR-10 ($50K$ images, $32 \times 32$ pixels) (Krizhevsky & Hinton (2009)); 2) STL-10 ($100K$ images, $48 \times 48$ pixels) (Coates et al. (2011)); 3) CelebA (about $203K$ images, $64 \times 64$ pixels) (Liu et al. (2015)); and 4) LSUN bedrooms (around 3 million images, $64 \times 64$ pixels) (Yu et al. (2015)). The images were scaled to range $[-1, 1]$ to avoid numeric issues.

**Network architecture:** The DCGAN (Radford et al. (2016)) architecture was used with hyperparameters from Miyato et al. (2018) (see Appendix B.2 for details). In all experiments, batch normalization (BN) (Ioffe & Szegedy (2015)) was used in the generator, and spectral normalization with the generalized power iteration (see Appendix C) in the discriminator. For MMD related losses, the dimension of discriminator output layer was set to 16; for non-saturating loss and hinge loss, it was 1. In Appendix D.2, we investigate the impact of discriminator output dimension on the performance of repulsive loss.

---

[3]Note that training stability is different from the local stability considered in Appendix A. Training stability often refers to the ability of model converging to a desired state measured by some criterion. Local stability means that if a model is initialized sufficiently close to an equilibrium, it will converge to the equilibrium.

Table 1: Inception score (IS), Fréchet Inception distance (FID) and multi-scale structural similarity (MS-SSIM) on image generation tasks using different loss functions

| Methods[1] | CIFAR-10 | | STL-10 | | CelebA[2] | | LSUN-bedrom[2] | |
|---|---|---|---|---|---|---|---|---|
| | IS | FID | IS | FID | FID | MS-SSIM | FID | MS-SSIM |
| Real data | 11.31 | 2.09 | 26.37 | 2.10 | 1.09 | 0.2678 | 1.24 | 0.0915 |
| Non-saturating | 7.39 | 23.23 | 8.25 | 48.53 | 10.64 | 0.2895 | 23.66 | 0.1027 |
| Hinge | 7.33 | 23.46 | 8.24 | 49.44 | 8.60 | 0.2894 | 16.73 | 0.0946 |
| MMD-rbf[3] | 7.05 | 28.38 | 8.13 | 57.52 | 13.03 | 0.2937 | | |
| MMD-rq[3] | 7.22 | 27.00 | 8.11 | 54.05 | 12.74 | 0.2935 | | |
| MMD-rbf-b | 7.18 | 25.25 | 8.07 | 51.86 | 10.09 | 0.3090 | 32.29 | 0.1001 |
| MMD-rep | 7.99 | **16.65** | **9.36** | **36.67** | 7.20 | 0.2761 | 16.91 | **0.0901** |
| MMD-rep-b | **8.29** | **16.21** | 9.34 | 37.63 | **6.79** | **0.2659** | **12.52** | **0.0908** |

[1] The models here differ only by the loss functions and dimension of discriminator outputs. See Section 5.1.
[2] For CelebA and LSUN-bedroom, IS is not meaningful (Bińkowski et al. (2018)) and thus omitted.
[3] On LSUN-bedroom, MMD-rbf and MMD-rq did not achieve reasonable results and thus are omitted.

**Hyper-parameters:** We used Adam optimizer (Kingma & Ba (2015)) with momentum parameters $\beta_1 = 0.5$, $\beta_2 = 0.999$; two-timescale update rule (TTUR) (Heusel et al. (2017)) with two learning rates $(\rho_D, \rho_G)$ chosen from $\{1e\text{-}4, 2e\text{-}4, 5e\text{-}4, 1e\text{-}3\}$ (16 combinations in total); and batch size $64$. Fine-tuning on learning rates may improve the model performance, but constant learning rates were used for simplicity. All models were trained for $100K$ iterations on CIFAR-10, STL-10, CelebA and LSUN bedroom datasets, with $n_{dis} = 1$, i.e., one discriminator update per generator update[4]. For MMD-rbf, the kernel scales $\sigma_i \in \{1, \sqrt{2}, 2, 2\sqrt{2}, 4\}$ were used due to a better performance than the original values used in Li et al. (2017a). For MMD-rq, $\alpha_i \in \{0.2, 0.5, 1, 2, 5\}$. For MMD-rbf-b, MMD-rep, MMD-rep-b, a single Gaussian kernel with $\sigma = 1$ was used.

**Evaluation metrics:** Inception score (IS) (Salimans et al. (2016)), Fréchet Inception distance (FID) (Heusel et al. (2017)) and multi-scale structural similarity (MS-SSIM) (Wang et al. (2003)) were used for quantitative evaluation. Both IS and FID are calculated using a pre-trained Inception model (Szegedy et al. (2016)). Higher IS and lower FID scores indicate better image quality. MS-SSIM calculates the pair-wise image similarity and is used to detect mode collapses among images of the same class (Odena et al. (2017)). Lower MS-SSIM values indicate perceptually more diverse images. For each model, $50K$ randomly generated samples and $50K$ real samples were used to calculate IS, FID and MS-SSIM.

## 5.2 QUANTITATIVE ANALYSIS

Table 1 shows the Inception score, FID and MS-SSIM of applying different loss functions on the benchmark datasets with the optimal learning rate combinations tested experimentally. Note that the same training setup (i.e., DCGAN + BN + SN + TTUR) was applied for each loss function. We observed that: 1) MMD-rep and MMD-rep-b performed significantly better than MMD-rbf and MMD-rbf-b respectively, showing the proposed repulsive loss $L_D^{\text{rep}}$ (Eq. 4) greatly improved over the attractive loss $L_D^{\text{att}}$ (Eq. 3); 2) Using a single kernel, MMD-rbf-b performed better than MMD-rbf and MMD-rq which used a linear combination of five kernels, indicating that the kernel saturation may be an issue that slows down MMD-GAN training; 3) MMD-rep-b performed comparable or better than MMD-rep on benchmark datasets where we found the RBF-B kernel managed to stabilize MMD-GAN training using repulsive loss. 4) MMD-rep and MMD-rep-b performed significantly better than the non-saturating and hinge losses, showing the efficacy of the proposed repulsive loss.

Additionally, we trained MMD-GAN using the general loss $L_{D,\lambda}$ (Eq. 5) for discriminator and $L_G^{\text{mmd}}$ (Eq. 2) for generator on the CIFAR-10 dataset. Fig. 3 shows the influence of $\lambda$ on the performance

---

[4]Increasing or decreasing $n_{dis}$ may improve the model performance in some cases, but it has significant impact on the computation cost. For simple and fair comparison, we set $n_{dis} = 1$ in all cases.

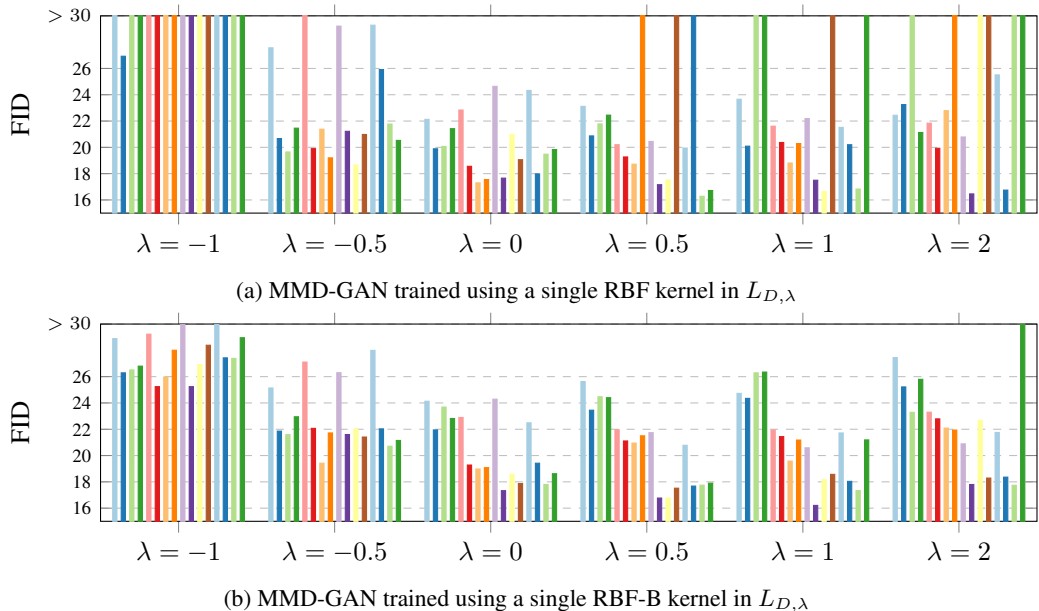

(a) MMD-GAN trained using a single RBF kernel in $L_{D,\lambda}$

(b) MMD-GAN trained using a single RBF-B kernel in $L_{D,\lambda}$

Figure 3: FID scores of MMD-GAN using (a) RBF kernel and (b) RBF-B kernel in $L_{D,\lambda}$ on CIFAR-10 dataset for 16 learning rate combinations. Each color bar represents the FID score using a learning rate combination $(\rho_D, \rho_G)$, in the order of $(1e\text{-}4, 1e\text{-}4)$, $(1e\text{-}4, 2e\text{-}4)$,...,$(1e\text{-}3, 1e\text{-}3)$. The discriminator was trained using $L_{D,\lambda}$ (Eq. 5) with $\lambda \in \{\text{-}1, \text{-}0.5, 0, 0.5, 1, 2\}$, and generator using $L_G^{\mathrm{mmd}}$ (Eq. 2). We use the FID$> 30$ to indicate that the model diverged or produced poor results.

of MMD-GAN with RBF and RBF-B kernel[5]. Note that when $\lambda = -1$, the models are essentially MMD-rbf (with a single Gaussian kernel) and MMD-rbf-b when RBF and RBF-B kernel are used respectively. We observed that: 1) the model performed well using repulsive loss (i.e., $\lambda \geq 0$), with $\lambda = 0.5, 1$ slightly better than $\lambda = -0.5, 0, 2$; 2) the MMD-rbf model can be significantly improved by simply increasing $\lambda$ from $-1$ to $-0.5$, which reduces the attraction of discriminator on real sample scores; 3) larger $\lambda$ may lead to more diverged models, possibly because the discriminator focuses more on expanding the real sample scores over adversarial learning; note when $\lambda \gg 1$, the model would simply learn to expand all real sample scores and pull the generated sample scores to real samples', which is a divergent process; 4) the RBF-B kernel managed to stabilize MMD-rep for most diverged cases but may occasionally cause the FID score to rise up.

The proposed methods were further evaluated in Appendix A, C and D. In Appendix A.2, we used a simulation study to show the local stability of MMD-rep trained by gradient descent, while its global stability is not guaranteed as bad initialization may lead to trivial solutions. The problem may be alleviated by adjusting the learning rate for generator. In Appendix C.3, we showed the proposed generalized power iteration (Section 4.2) imposes a stronger Lipschitz constraint than the method in Miyato et al. (2018), and benefited MMD-GAN training using the repulsive loss. Moreover, the RBF-B kernel managed to stabilize the MMD-GAN training for various configurations of the spectral normalization method. In Appendix D.1, we showed the gradient penalty can also be used with the repulsive loss. In Appendix D.2, we showed that it was better to use more than one neuron at the discriminator output layer for the repulsive loss.

### 5.3 QUALITATIVE ANALYSIS

The discriminator outputs may be interpreted as a learned representation of the input samples. Fig. 4 visualizes the discriminator outputs learned by the MMD-rbf and proposed MMD-rep methods on CIFAR-10 dataset using t-SNE (van der Maaten (2014)). MMD-rbf ignored the class structure in data (see Fig. 4a) while MMD-rep learned to concentrate the data from the same class and separate different classes to some extent (Fig. 4b). This is because the discriminator $D$ has to actively learn

---

[5]For $\lambda < 0$, the RBF-B kernel in Eq. 6 was used in $L_{D,\lambda}$.

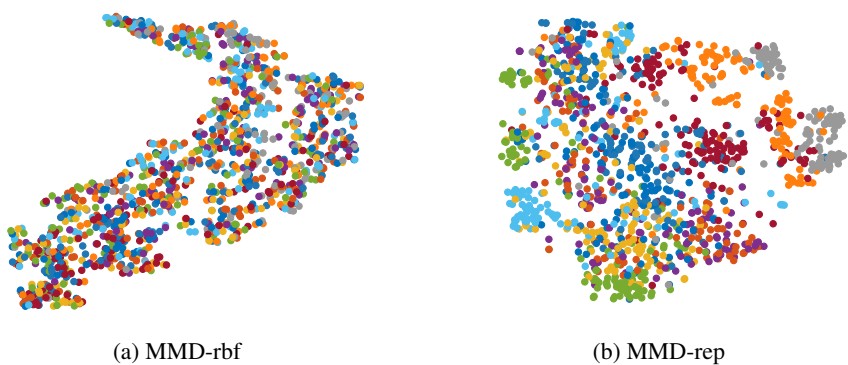

(a) MMD-rbf                                      (b) MMD-rep

Figure 4: t-SNE visualization of discriminator outputs $\{D(\boldsymbol{x})\}$ learned by (a) MMD-rbf and (b) MMD-rep for 2560 real samples from the CIFAR-10 dataset, colored by their class labels.

the data structure in order to expands the real sample scores $\{D(\boldsymbol{x})\}$. Thus, we speculate that techniques reinforcing the learning of cluster structures in data may further improve the training of MMD-GAN.

In addition, the performance gain of proposed repulsive loss (Eq. 4) over the attractive loss (Eq. 3) comes at no additional computational cost. In fact, by using a single kernel rather than a linear combination of kernels, MMD-rep and MMD-rep-b are simpler than MMD-rbf and MMD-rq. Besides, given a typically small batch size and a small number of discriminator output neurons (64 and 16 in our experiments), the cost of MMD over the non-saturating and hinge loss is marginal compared to the convolution operations.

In Appendix D.3, we provide some random samples generated by the methods in our study.

## 6  DISCUSSION

This study extends the previous work on MMD-GAN (Li et al. (2017a)) with two contributions. First, we interpreted the optimization of MMD loss as a combination of attraction and repulsion processes, and proposed a repulsive loss for the discriminator that actively learns the difference among real data. Second, we proposed a bounded Gaussian RBF (RBF-B) kernel to address the saturation issue. Empirically, we observed that the repulsive loss may result in unstable training, due to factors including initialization (Appendix A.2), learning rate (Fig. 3b) and Lipschitz constraints on the discriminator (Appendix C.3). The RBF-B kernel managed to stabilize the MMD-GAN training in many cases. Tuning the hyper-parameters in RBF-B kernel or using other regularization methods may further improve our results.

The theoretical advantages of MMD-GAN require the discriminator to be injective. The proposed repulsive loss (Eq. 4) attempts to realize this by explicitly maximizing the pair-wise distances among the real samples. Li et al. (2017a) achieved the injection property by using the discriminator as the encoder and an auxiliary network as the decoder to reconstruct the real and generated samples, which is more computationally extensive than our proposed approach. On the other hand, Bińkowski et al. (2018); Arbel et al. (2018) imposed a Lipschitz constraint on the discriminator in MMD-GAN via gradient penalty, which may not necessarily promote an injective discriminator.

The idea of repulsion on real sample scores is in line with existing studies. It has been widely accepted that the quality of generated samples can be significantly improved by integrating labels (Odena et al. (2017); Miyato & Koyama (2018); Zhou et al. (2018)) or even pseudo-labels generated by k-means method (Grinblat et al. (2017)) in the training of discriminator. The reason may be that the labels help concentrate the data from the same class and separate those from different classes. Using a pre-trained classifier may also help produce vivid image samples (Huang et al. (2017)) as the learned representations of the real samples in the hidden layers of the classifier tend to be well separated/organized and may produce more meaningful gradients to the generator.

At last, we note that the proposed repulsive loss is orthogonal to the GAN studies on designing network structures and training procedures, and thus may be combined with a variety of novel techniques. For example, the ResNet architecture (He et al. (2016)) has been reported to outperform the plain DCGAN used in our experiments on image generation tasks (Miyato et al. (2018); Gulrajani et al. (2017)) and self-attention module may further improve the results (Zhang et al. (2018)). On the other hand, Karras et al. (2018) proposed to progressively grows the size of both discriminator and generator and achieved the state-of-the-art performance on unsupervised training of GANs on the CIFAR-10 dataset. Future work may explore these directions.

ACKNOWLEDGMENTS

Wei Wang is fully supported by the Ph.D. scholarships of The University of Melbourne. This work is partially funded by Australian Research Council grant DP150103512 and undertaken using the LIEF HPC-GPGPU Facility hosted at the University of Melbourne. The Facility was established with the assistance of LIEF Grant LE170100200.

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

# Appendices

## A   STABILITY ANALYSIS OF MMD-GAN

This section demonstrates that, under mild assumptions, MMD-GAN trained by gradient descent is locally exponentially stable at equilibrium. It is organized as follows. The main assumption and proposition are presented in Section A.1, followed by simulation study in Section A.2 and proof in Section A.3. We discuss the indications of assumptions on the discriminator of GAN in Section A.4.

### A.1   MAIN PROPOSITION

We consider GAN trained using the MMD loss $L_G^{\text{mmd}}$ for generator $G$ and either the attractive loss $L_D^{\text{att}}$ or repulsive loss $L_D^{\text{rep}}$ for discriminator $D$, listed below:

$$L_G^{\text{mmd}} = M_{k \circ D}^2(P_{\mathbf{X}}, P_G) = \mathbb{E}_{P_{\mathbf{X}}}[k_D(\boldsymbol{x}, \boldsymbol{x}')] - 2\mathbb{E}_{P_{\mathbf{X}}, P_G}[k_D(\boldsymbol{x}, \boldsymbol{y})] + \mathbb{E}_{P_G}[k_D(\boldsymbol{y}, \boldsymbol{y}')] \quad \text{(S1a)}$$

$$L_D^{\text{att}} = -L_G^{\text{mmd}} \quad \text{(S1b)}$$

$$L_D^{\text{rep}} = \mathbb{E}_{P_{\mathbf{X}}}[k_D(\boldsymbol{x}, \boldsymbol{x}')] - \mathbb{E}_{P_G}[k_D(\boldsymbol{y}, \boldsymbol{y}')] \quad \text{(S1c)}$$

where $k_D(\boldsymbol{a}, \boldsymbol{b}) = k(D(\boldsymbol{a}), D(\boldsymbol{b}))$. Let $\mathcal{S}(P)$ be the support of distribution $P$; let $\boldsymbol{\theta}_G \in \Theta_G$, $\boldsymbol{\theta}_D \in \Theta_D$ be the parameters of the generator $G$ and discriminator $D$ respectively. To prove that GANs trained using the minimax loss and gradient updates is locally stable at the equilibrium point $(\boldsymbol{\theta}_D^*, \boldsymbol{\theta}_G^*)$, Nagarajan & Kolter (2017) made the following assumption:

**Assumption 1** (Nagarajan & Kolter (2017))**.** $P_{\boldsymbol{\theta}_G^*} = P_{\mathbf{X}}$ and $\forall \boldsymbol{x} \in \mathcal{S}(P_{\mathbf{X}}), D_{\boldsymbol{\theta}_D^*}(\boldsymbol{x}) = 0$.

For loss functions like minimax and Wasserstein, $D_{\boldsymbol{\theta}_D}(\boldsymbol{x})$ may be interpreted as how plausible a sample is real. Thus at equilibrium, it may be reasonable to assume all real and generated samples are equally plausible. However, $D_{\boldsymbol{\theta}_D^*}(\boldsymbol{x}) = 0$ also indicates that $D_{\boldsymbol{\theta}_D^*}$ may have no discrimination power (see Appendix A.4 for discussion). For MMD loss in Eq. S1, $D_{\boldsymbol{\theta}_D}(\boldsymbol{x})|_{\boldsymbol{x} \sim P}$ may be interpreted as a learned representation of the distribution $P$. As long as two distributions $P$ and $Q$ match, $M_{k \circ D_{\boldsymbol{\theta}_D}}^2(P, Q) = 0$. On the other hand, $D_{\boldsymbol{\theta}_D}(\boldsymbol{x}) = 0$ is a minima solution for $D$ but $D$ is trained to find local maxima. Thus in contrast to Assumption 1, we assume

**Assumption 2.** *For GANs using MMD loss in Eq. S1, and random initialization on parameters, at equilibrium, $D_{\boldsymbol{\theta}_D^*}(\boldsymbol{x})$ is injective on $\mathcal{S}(P_{\mathbf{X}}) \bigcup \mathcal{S}(P_{\boldsymbol{\theta}_G^*})$.*

Assumption 2 indicates that $D_{\boldsymbol{\theta}_D^*}(\boldsymbol{x})$ is not constant almost everywhere. We use a simulation study in Section A.2 to show that $D_{\boldsymbol{\theta}_D^*}(\boldsymbol{x}) = 0$ does not hold in general for MMD loss. Based on Assumption 2, we propose the following proposition and prove it in Appendix A.3:

**Proposition 1.** *If there exists $\boldsymbol{\theta}_G^* \in \Theta_G$ such that $P_{\boldsymbol{\theta}_G^*} = P_{\mathbf{X}}$, then GANs with MMD loss in Eq. S1 has equilibria $(\boldsymbol{\theta}_G^*, \boldsymbol{\theta}_D)$ for any $\boldsymbol{\theta}_D \in \Theta_D$. Moreover, the model trained using gradient descent methods is locally exponentially stable at $(\boldsymbol{\theta}_G^*, \boldsymbol{\theta}_D)$ for any $\boldsymbol{\theta}_D \in \Theta_D$.*

There may exist non-realizable cases where the mapping between $P_{\mathbf{Z}}$ and $P_{\mathbf{X}}$ cannot be represented by any generator $G_{\boldsymbol{\theta}_G}$ with $\boldsymbol{\theta}_G \in \Theta_G$. In Section A.2, we use a simulation study to show that both the attractive MMD loss $L_D^{\text{att}}$ (Eq. S1b) and the proposed repulsive loss $L_D^{\text{rep}}$ (Eq. S1c) may be locally stable and leave the proof for future work.

### A.2   SIMULATION STUDY

In this section, we reused the example from Nagarajan & Kolter (2017) to show that GAN trained using the MMD loss in Eq. S1 is locally stable. Consider a two-parameter MMD-GAN with uniform latent distribution $P_Z$ over $[-1, 1]$, generator $G(z) = w_1 z$, discriminator $D(x) = w_2 x^2$, and Gaussian kernel $k_{0.5}^{\text{rbf}}$. The MMD-rbf model ($L_G^{\text{mmd}}$ and $L_D^{\text{att}}$ from Eq. S1b) and the MMD-rep model ($L_G^{\text{mmd}}$ and $L_D^{\text{rep}}$ from Eq. S1c) were tested. Each model was applied to two cases:

 (a)  the data distribution $P_X$ is the same as $P_Z$, i.e., uniform over $[-1, 1]$, thus $P_X$ is realizable;

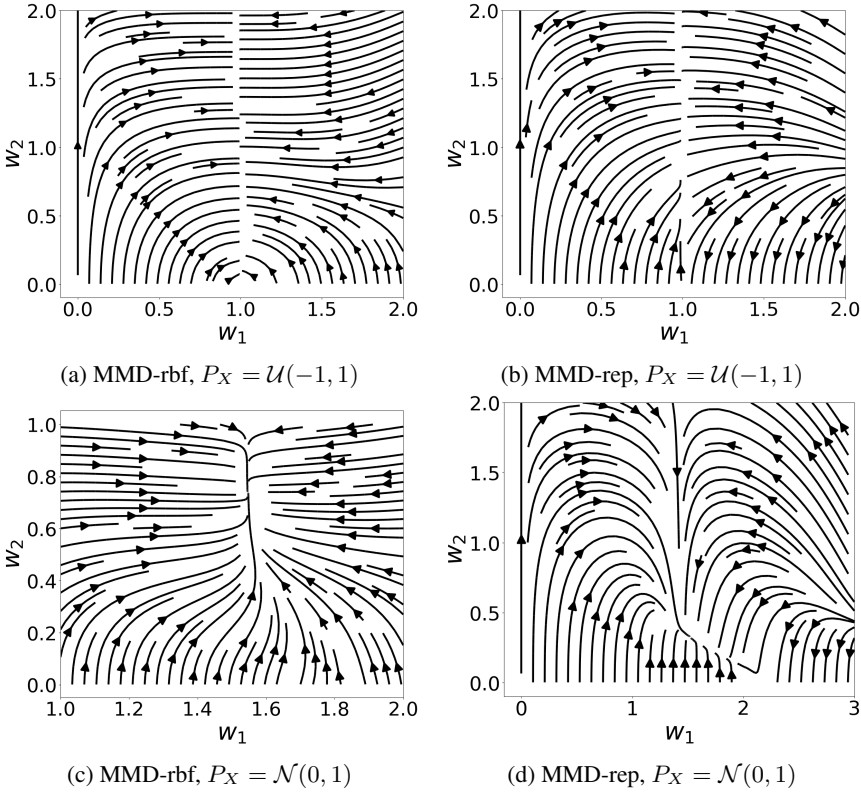

Figure S1: Streamline plots of MMD-GAN using the MMD-rbf and the MMD-rep model on distributions: $P_Z = \mathcal{U}(-1,1)$, $P_X = \mathcal{U}(-1,1)$ or $P_X = \mathcal{N}(0,1)$. In (a) and (b), the equilibria satisfying $P_G = P_X$ lie on the line $w_1 = 1$. In (c), the equilibrium lies around point $(1.55, 0.74)$; in (d), it is around $(1.55, 0.32)$.

(b) $P_X$ is standard Gaussian, thus non-realizable for any $w_1 \in \mathbb{R}$.

Fig. S1 shows that MMD-GAN are locally stable in both cases and $D_{\boldsymbol{\theta}_D^*}(\boldsymbol{x}) = 0$ does not hold in general for MMD loss. However, MMD-rep may not be globally stable for the tested cases: initialization of $(w_1, w_2)$ in some regions may lead to the trivial solution $w_2 = 0$ (see Fig. S1b and S1d). We note that by decreasing the learning rate for $G$, the area of such regions decreased. At last, it is interesting to note that both MMD-rbf and MMD-rep had the same nontrivial solution $w_1 \approx 1.55$ for generator in the non-realizable cases (see Fig. S1c and S1d).

### A.3 PROOF OF PROPOSITION 1

This section divides the proof for Proposition 1 into two parts. First, we show that GAN with the MMD loss in Eq. S1 has equilibria for any parameter configuration of discriminator $D$; second, we prove the model is locally exponentially stable. For convenience, we consider the general form of discriminator loss in Eq. 5:

$$L_{D,\lambda} = \lambda \mathbb{E}_{P_{\mathbf{X}}}[k_D(\boldsymbol{x}, \boldsymbol{x}')] - (\lambda - 1)\mathbb{E}_{P_{\mathbf{X}}, P_G}[k_D(\boldsymbol{x}, \boldsymbol{y})] - \mathbb{E}_{P_G}[k_D(\boldsymbol{y}, \boldsymbol{y}')] \qquad \text{(S2)}$$

which has $L_D^{\text{att}}$ and $L_D^{\text{rep}}$ as the special cases when $\lambda$ equals $-1$ and $1$ respectively. Consider real data $\mathbf{X}_r \sim P_{\mathbf{X}}$, latent variable $\mathbf{Z} \sim P_{\mathbf{Z}}$ and generated variable $\mathbf{Y}_g = G_{\boldsymbol{\theta}_G}(\mathbf{Z})$. Let $\boldsymbol{x}_r, \boldsymbol{z}, \boldsymbol{y}_g$ be their samples. Denote $\nabla_{\boldsymbol{b}}^{\boldsymbol{a}} = \frac{\partial \boldsymbol{a}}{\partial \boldsymbol{b}}$; $\dot{\boldsymbol{\theta}}_D = -\nabla_{\boldsymbol{\theta}_D}^{L_D}$, $\dot{\boldsymbol{\theta}}_G = -\nabla_{\boldsymbol{\theta}_G}^{L_G}$; $\boldsymbol{d}_g = D(G(\boldsymbol{z}))$, $\boldsymbol{d}_r = D(\boldsymbol{x}_r)$ where $L_D$ and $L_G$ are the losses for $D$ and $G$ respectively. Assume an isotropic stationary kernel $k(\boldsymbol{a}, \boldsymbol{b}) = k_I(\|\boldsymbol{a} - \boldsymbol{b}\|)$ (Genton (2002)) is used in MMD. We first show:

**Proposition 1** (Part 1). *If there exists $\boldsymbol{\theta}_G^* \in \Theta_G$ such that $P_{\boldsymbol{\theta}_G^*} = P_{\mathbf{X}}$, the GAN with the MMD loss in Eq. S1a and Eq. S2 has equilibria $(\boldsymbol{\theta}_G^*, \boldsymbol{\theta}_D)$ for any $\boldsymbol{\theta}_D \in \Theta_D$.*

*Proof.* Denote $\boldsymbol{e}_{i,j} = \boldsymbol{a}_i - \boldsymbol{b}_j$ and $\nabla^k_{\boldsymbol{e}_{i,j}} = \frac{\partial k(\boldsymbol{a}_i, \boldsymbol{b}_j)}{\partial \boldsymbol{e}}$ where $k$ is the kernel of MMD. The gradients of MMD loss are

$$\dot{\boldsymbol{\theta}}_D = (\lambda - 1)\mathbb{E}_{P_{\mathbf{X}}, P_{\boldsymbol{\theta}_G}}[\nabla^k_{\boldsymbol{e}_{r,g}} \nabla^{\boldsymbol{e}_{r,g}}_{\boldsymbol{\theta}_D}] - \lambda \mathbb{E}_{P_{\mathbf{X}}}[\nabla^k_{\boldsymbol{e}_{r1,r2}} \nabla^{\boldsymbol{e}_{r1,r2}}_{\boldsymbol{\theta}_D}] + \mathbb{E}_{P_{\boldsymbol{\theta}_G}}[\nabla^k_{\boldsymbol{e}_{g1,g2}} \nabla^{\boldsymbol{e}_{g1,g2}}_{\boldsymbol{\theta}_D}] \quad \text{(S3a)}$$

$$\dot{\boldsymbol{\theta}}_G = 2\mathbb{E}_{P_{\boldsymbol{\theta}_G}, P_{\mathbf{X}}}[\nabla^k_{\boldsymbol{e}_{g,r}} \nabla^{\boldsymbol{d}_g}_{\boldsymbol{x}_g} \nabla^{\boldsymbol{x}_g}_{\boldsymbol{\theta}_G}] - \mathbb{E}_{P_{\boldsymbol{\theta}_G}}[\nabla^k_{\boldsymbol{e}_{g1,g2}}(\nabla^{\boldsymbol{d}_{g1}}_{\boldsymbol{x}_{g1}} \nabla^{\boldsymbol{x}_{g1}}_{\boldsymbol{\theta}_G} - \nabla^{\boldsymbol{d}_{g2}}_{\boldsymbol{x}_{g2}} \nabla^{\boldsymbol{x}_{g2}}_{\boldsymbol{\theta}_G})] \quad \text{(S3b)}$$

Note that, given i.i.d. drawn samples $\boldsymbol{X} = \{\boldsymbol{x}_i\}_{i=1}^n \sim P_{\mathbf{X}}$ and $\boldsymbol{Y} = \{\boldsymbol{y}_i\}_{i=1}^n \sim P_G$, an unbiased estimator of the squared MMD is (Gretton et al. (2012))

$$\hat{M}_k^2(P_{\mathbf{X}}, P_G) = \frac{1}{n(n-1)} \sum_{i \neq j}^n k(\boldsymbol{x}_i, \boldsymbol{x}_j) + \frac{1}{n(n-1)} \sum_{i \neq j}^n k(\boldsymbol{y}_i, \boldsymbol{y}_j) - \frac{2}{n(n-1)} \sum_{i \neq j}^n k(\boldsymbol{x}_i, \boldsymbol{y}_j) \quad \text{(S4)}$$

At equilibrium, consider a sequence of $N$ samples $\boldsymbol{d}_{ri} = \boldsymbol{d}_{gi} = \boldsymbol{d}_i$ with $N \to \infty$, we have

$$\dot{\boldsymbol{\theta}}_D \propto (\lambda - 1)\sum_{i \neq j} \nabla^k_{\boldsymbol{e}_{i,j}} \nabla^{\boldsymbol{e}_{i,j}}_{\boldsymbol{\theta}_D} - \lambda \sum_{i \neq j} \nabla^k_{\boldsymbol{e}_{i,j}} \nabla^{\boldsymbol{e}_{i,j}}_{\boldsymbol{\theta}_D} + \sum_{i \neq j} \nabla^k_{\boldsymbol{e}_{i,j}} \nabla^{\boldsymbol{e}_{i,j}}_{\boldsymbol{\theta}_D} = \boldsymbol{0}$$

$$\dot{\boldsymbol{\theta}}_G^* \propto -\sum_{i \neq j} \nabla^k_{\boldsymbol{e}_{i,j}}(\nabla^{\boldsymbol{d}_i}_{\boldsymbol{x}_i} \nabla^{\boldsymbol{x}_i}_{\boldsymbol{\theta}_G} - \nabla^{\boldsymbol{d}_j}_{\boldsymbol{x}_j} \nabla^{\boldsymbol{x}_j}_{\boldsymbol{\theta}_G}) + 2\sum_{i \neq j} \nabla^k_{\boldsymbol{e}_{i,j}} \nabla^{\boldsymbol{d}_i}_{\boldsymbol{x}_i} \nabla^{\boldsymbol{x}_i}_{\boldsymbol{\theta}_G}$$

$$= \sum_{i \neq j} \nabla^k_{\boldsymbol{e}_{i,j}}(\nabla^{\boldsymbol{d}_i}_{\boldsymbol{x}_i} \nabla^{\boldsymbol{x}_i}_{\boldsymbol{\theta}_G} + \nabla^{\boldsymbol{d}_j}_{\boldsymbol{x}_j} \nabla^{\boldsymbol{x}_j}_{\boldsymbol{\theta}_G}) = \boldsymbol{0}$$

where for $\dot{\boldsymbol{\theta}}_G^*$ we have used the fact that for each term in the summation, there exists an term with $i, j$ reversed and $\nabla^k_{\boldsymbol{e}_{i,j}} = -\nabla^k_{\boldsymbol{e}_{j,i}}$ thus the summation is zero. Since we have not assumed the status of $\boldsymbol{\theta}_D$, $\dot{\boldsymbol{\theta}}_D = \boldsymbol{0}$ for any $\boldsymbol{\theta}_D \in \Theta_D$. $\qquad \square$

We proceed to prove the model stability. First, following Theorem 5 in Gretton et al. (2012) and Theorem 4 in Li et al. (2017a), it is straightforward to see:

**Lemma A.1.** *Under Assumption 2, $M^2_{k \circ D_{\boldsymbol{\theta}_D}}(P_{\mathbf{X}}, P_{\boldsymbol{\theta}_G}) \geq 0$ with the equality if and only if $P_{\mathbf{X}} = P_{\boldsymbol{\theta}_G}$.*

Lemma A.1 and Proposition 1 (Part 1) state that at equilibrium $P_{\boldsymbol{\theta}_G^*} = P_{\mathbf{X}}$, every discriminator $D_{\boldsymbol{\theta}_D}$ and kernel $k$ will give $M^2_{k \circ D_{\boldsymbol{\theta}_D}}(P_{\boldsymbol{\theta}_G^*}, P_{\mathbf{X}}) = 0$, thus no discriminator can distinguish the two distributions. On the other hand, we cite Theorem A.4 from Nagarajan & Kolter (2017):

**Lemma A.2** (Nagarajan & Kolter (2017))**.** *Consider a non-linear system of parameters $(\boldsymbol{\theta}, \boldsymbol{\gamma})$: $\dot{\boldsymbol{\theta}} = h_1(\boldsymbol{\theta}, \boldsymbol{\gamma})$, $\dot{\boldsymbol{\gamma}} = h_2(\boldsymbol{\theta}, \boldsymbol{\gamma})$ with an equilibrium point at $(\boldsymbol{0}, \boldsymbol{0})$. Let there exist $\epsilon$ such that $\forall \boldsymbol{\gamma} \in \mathbb{B}_\epsilon(\boldsymbol{0})$, $(\boldsymbol{0}, \boldsymbol{\gamma})$ is an equilibrium. If $\boldsymbol{J} = \frac{\partial h_1(\boldsymbol{\theta}, \boldsymbol{\gamma})}{\partial \boldsymbol{\theta}}\big|_{(\boldsymbol{0}, \boldsymbol{0})}$ is a Hurwitz matrix, the non-linear system is exponentially stable.*

Now we can prove:

**Proposition 1** (Part 2)**.** *At equilibrium $P_{\boldsymbol{\theta}_G^*} = P_{\mathbf{X}}$, the GAN trained using MMD loss and gradient descent methods is locally exponentially stable at $(\boldsymbol{\theta}_G^*, \boldsymbol{\theta}_D)$ for any $\boldsymbol{\theta}_D \in \Theta_D$.*

*Proof.* Inspired by Nagarajan & Kolter (2017), we first derive the Jacobian of the system

$$\boldsymbol{J} = \begin{bmatrix} \boldsymbol{J}_{DD} & \boldsymbol{J}_{DG} \\ \boldsymbol{J}_{GD} & \boldsymbol{J}_{GG} \end{bmatrix} \triangleq \begin{bmatrix} \partial \dot{\boldsymbol{\theta}}_D^T / \partial \boldsymbol{\theta}_D & \partial \dot{\boldsymbol{\theta}}_D^T / \partial \boldsymbol{\theta}_G \\ \partial \dot{\boldsymbol{\theta}}_G^T / \partial \boldsymbol{\theta}_D & \partial \dot{\boldsymbol{\theta}}_G^T / \partial \boldsymbol{\theta}_G \end{bmatrix}$$

Denote $\Delta_{\boldsymbol{b}}^{\boldsymbol{a}} = \frac{\partial^2 \boldsymbol{a}}{\partial \boldsymbol{b}^2}$ and $\Delta_{\boldsymbol{bc}}^{\boldsymbol{a}} = \frac{\partial^2 \boldsymbol{a}}{\partial \boldsymbol{b} \partial \boldsymbol{c}}$. Based on Eq. S3, we have

$$\boldsymbol{J}_{DD} = (\lambda - 1)\mathbb{E}_{P_{\mathbf{X}}, P_{\boldsymbol{\theta}_G}}[(\Delta_{\boldsymbol{\theta}_D}^{\boldsymbol{e}_{r,g}})^T \otimes (\nabla_{\boldsymbol{e}_{r,g}}^k)^T + (\nabla_{\boldsymbol{\theta}_D}^{\boldsymbol{e}_{r,g}})^T \Delta_{\boldsymbol{e}_{r,g}}^k \nabla_{\boldsymbol{\theta}_D}^{\boldsymbol{e}_{r,g}}] \tag{S5a}$$
$$- \lambda\mathbb{E}_{P_{\mathbf{X}}}[(\Delta_{\boldsymbol{\theta}_D}^{\boldsymbol{e}_{r1,r2}})^T \otimes (\nabla_{\boldsymbol{e}_{r1,r2}}^k)^T + (\nabla_{\boldsymbol{\theta}_D}^{\boldsymbol{e}_{r1,r2}})^T \Delta_{\boldsymbol{e}_{r1,r2}}^k \nabla_{\boldsymbol{\theta}_D}^{\boldsymbol{e}_{r1,r2}}]$$
$$+ \mathbb{E}_{P_{\boldsymbol{\theta}_G}}[(\Delta_{\boldsymbol{\theta}_D}^{\boldsymbol{e}_{g1,g2}})^T \otimes (\nabla_{\boldsymbol{e}_{g1,g2}}^k)^T + (\nabla_{\boldsymbol{\theta}_D}^{\boldsymbol{e}_{g1,g2}})^T \Delta_{\boldsymbol{e}_{g1,g2}}^k \nabla_{\boldsymbol{\theta}_D}^{\boldsymbol{e}_{g1,g2}}]$$

$$\boldsymbol{J}_{DG} = (\lambda - 1)\mathbb{E}_{P_{\mathbf{X}}, P_{\boldsymbol{\theta}_G}}[(\Delta_{\boldsymbol{\theta}_D \boldsymbol{\theta}_G}^{\boldsymbol{e}_{r,g}})^T \otimes (\nabla_{\boldsymbol{e}_{r,g}}^k)^T - (\nabla_{\boldsymbol{\theta}_D}^{\boldsymbol{e}_{r,g}})^T \Delta_{\boldsymbol{e}_{r,g}}^k \nabla_{\boldsymbol{\theta}_G}^{\boldsymbol{d}_g}]$$
$$+ \mathbb{E}_{P_{\boldsymbol{\theta}_G}}[(\Delta_{\boldsymbol{\theta}_D \boldsymbol{\theta}_G}^{\boldsymbol{e}_{g1,g2}})^T \otimes (\nabla_{\boldsymbol{e}_{g1,g2}}^k)^T + (\nabla_{\boldsymbol{\theta}_D}^{\boldsymbol{e}_{g1,g2}})^T \Delta_{\boldsymbol{e}_{g1,g2}}^k \nabla_{\boldsymbol{\theta}_G}^{\boldsymbol{e}_{g1,g2}}] \tag{S5b}$$

$$\boldsymbol{J}_{GD} = 2\mathbb{E}_{P_{\mathbf{X}}, P_{\boldsymbol{\theta}_G}}[(\Delta_{\boldsymbol{\theta}_G \boldsymbol{\theta}_D}^{\boldsymbol{e}_{g,r}})^T \otimes (\nabla_{\boldsymbol{e}_{g,r}}^k)^T + (\nabla_{\boldsymbol{\theta}_G}^{\boldsymbol{e}_{g,r}})^T \Delta_{\boldsymbol{e}_{g,r}}^k \nabla_{\boldsymbol{\theta}_D}^{\boldsymbol{d}_g}]$$
$$- \mathbb{E}_{P_{\boldsymbol{\theta}_G}}[(\Delta_{\boldsymbol{\theta}_G \boldsymbol{\theta}_D}^{\boldsymbol{e}_{g1,g2}})^T \otimes (\nabla_{\boldsymbol{e}_{g1,g2}}^k)^T + (\nabla_{\boldsymbol{\theta}_G}^{\boldsymbol{e}_{g1,g2}})^T \Delta_{\boldsymbol{e}_{g1,g2}}^k \nabla_{\boldsymbol{\theta}_D}^{\boldsymbol{e}_{g1,g2}}] \tag{S5c}$$

$$\boldsymbol{J}_{GG} = - \mathbb{E}_{P_{\boldsymbol{\theta}_G}}[(\Delta_{\boldsymbol{\theta}_G}^{\boldsymbol{e}_{g1,g2}})^T \otimes (\nabla_{\boldsymbol{e}_{g1,g2}}^k)^T + (\nabla_{\boldsymbol{\theta}_G}^{\boldsymbol{e}_{g1,g2}})^T \Delta_{\boldsymbol{e}_{g1,g2}}^k \nabla_{\boldsymbol{\theta}_G}^{\boldsymbol{e}_{g1,g2}}]$$
$$+ 2\mathbb{E}_{P_{\mathbf{X}}, P_{\boldsymbol{\theta}_G}}[(\Delta_{\boldsymbol{\theta}_G}^{\boldsymbol{d}_g})^T \otimes (\nabla_{\boldsymbol{e}_{r,g}}^k)^T + (\nabla_{\boldsymbol{\theta}_G}^{\boldsymbol{d}_g})^T \Delta_{\boldsymbol{e}_{r,g}}^k \nabla_{\boldsymbol{\theta}_G}^{\boldsymbol{d}_g}] \tag{S5d}$$

where $\otimes$ is the kronecker product. At equilibrium, consider a sequence of $N$ samples $\boldsymbol{d}_{ri} = \boldsymbol{d}_{gi} = \boldsymbol{d}_i$ with $N \to \infty$, we have $\boldsymbol{J}_{DD} = \boldsymbol{0}$, $\boldsymbol{J}_{GD} = \boldsymbol{0}$ and

$$\boldsymbol{J}_{DG} \propto (\lambda + 1) \sum_{i<j}[(\Delta_{\boldsymbol{\theta}_D \boldsymbol{\theta}_G}^{\boldsymbol{e}_{i,j}})^T \otimes (\nabla_{\boldsymbol{e}_{i,j}}^k)^T + (\nabla_{\boldsymbol{\theta}_D}^{\boldsymbol{e}_{i,j}})^T \Delta_{\boldsymbol{e}_{i,j}}^k \nabla_{\boldsymbol{\theta}_G}^{\boldsymbol{e}_{i,j}}]$$

$$\boldsymbol{J}_{GG} = \mathbb{E}_{P_{\boldsymbol{\theta}_G}}[(\nabla_{\boldsymbol{\theta}_G}^{\boldsymbol{d}_{g1}})^T \Delta_{\boldsymbol{e}_{g1,g2}}^k \nabla_{\boldsymbol{\theta}_G}^{\boldsymbol{d}_{g1}} + (\nabla_{\boldsymbol{\theta}_G}^{\boldsymbol{d}_{g2}})^T \Delta_{\boldsymbol{e}_{g1,g2}}^k \nabla_{\boldsymbol{\theta}_G}^{\boldsymbol{d}_{g2}} - (\nabla_{\boldsymbol{\theta}_G}^{\boldsymbol{e}_{g1,g2}})^T \Delta_{\boldsymbol{e}_{g1,g2}}^k \nabla_{\boldsymbol{\theta}_G}^{\boldsymbol{e}_{g1,g2}}]$$
$$= \mathbb{E}_{P_{\boldsymbol{\theta}_G}}[(\nabla_{\boldsymbol{\theta}_G}^{\boldsymbol{d}_{g1}})^T \Delta_{\boldsymbol{e}_{g1,g2}}^k \nabla_{\boldsymbol{\theta}_G}^{\boldsymbol{d}_{g2}} + (\nabla_{\boldsymbol{\theta}_G}^{\boldsymbol{d}_{g2}})^T \Delta_{\boldsymbol{e}_{g1,g2}}^k \nabla_{\boldsymbol{\theta}_G}^{\boldsymbol{d}_{g1}}]$$

Given Lemma A.1 and fact that $\boldsymbol{J}_{GG}$ is the Hessian matrix of $M_{k \circ D_{\boldsymbol{\theta}_D}}^2(P_{\mathbf{X}}, P_{\boldsymbol{\theta}_G})$, $\boldsymbol{J}_{GG}$ is negative semidefinite. The eigenvectors of $\boldsymbol{J}_{GG}$ corresponding to zero eigenvalues form $\mathrm{null}(\boldsymbol{J}_{GG})$. There may exist small distortion $\delta\boldsymbol{\theta}_G \in \mathrm{null}(\boldsymbol{J}_{GG})$ such that $P_{\boldsymbol{\theta}_G^* + \delta\boldsymbol{\theta}_G} = P_{\boldsymbol{\theta}_G^*}$. That is, $P_{\boldsymbol{\theta}_G^*}$ is locally constant along some directions in the parameter space of $G$. As a result, $\mathrm{null}(\boldsymbol{J}_{GG}) \subseteq \mathrm{null}(\boldsymbol{J}_{DG})$ because varying $\boldsymbol{\theta}_G^*$ along these directions has no effect on $D$.

Following Lemma C.3 of Nagarajan & Kolter (2017), we consider eigenvalue decomposition $\boldsymbol{J}_{GG} = \boldsymbol{U}_G \boldsymbol{\Lambda}_G \boldsymbol{U}_G^T$ and $\boldsymbol{J}_{DG} \boldsymbol{J}_{DG}^T = \boldsymbol{U}_D \boldsymbol{\Lambda}_D \boldsymbol{U}_D^T$. Let $\boldsymbol{U}_G = [\boldsymbol{T}_G'^T, \boldsymbol{T}_G^T]$, $\boldsymbol{U}_D = [\boldsymbol{T}_D'^T, \boldsymbol{T}_D^T]$ such that $\mathrm{Col}(\boldsymbol{T}_G'^T) = \mathrm{null}(\boldsymbol{J}_{GG})$, $\mathrm{Col}(\boldsymbol{T}_D'^T) = \mathrm{null}(\boldsymbol{J}_{DG}^T)$. Thus, the projections $\boldsymbol{\gamma}_G = \boldsymbol{T}_G \boldsymbol{\theta}_G$ are orthogonal to $\mathrm{null}(\boldsymbol{J}_{GG})$. Then, the Jacobian corresponding to the projected system has the form $\boldsymbol{J}' = [\boldsymbol{0}, \boldsymbol{J}_{DG}'; \boldsymbol{0}, \boldsymbol{J}_{GG}']$ with block $\boldsymbol{J}_{DG}' = \boldsymbol{T}_D \boldsymbol{J}_{DG} \boldsymbol{T}_G^T$ and $\boldsymbol{J}_{GG}' = \boldsymbol{T}_G \boldsymbol{J}_{GG} \boldsymbol{T}_G^T$, where $\boldsymbol{J}_{GG}'$ is negative definite. Moreover, on all directions exclude those described by $\boldsymbol{J}_{GG}'$, the system is surrounded by a neighborhood of equilibia at least locally. According to Lemma A.2, the system is exponentially stable. $\qquad\square$

### A.4 DISCUSSION ON ASSUMPTION 1

This section shows that constant discriminator output $D_{\boldsymbol{\theta}_D^*}(\boldsymbol{x}) = \boldsymbol{c}$ for $\boldsymbol{x} \in \mathcal{S}(P_{\mathbf{X}}) \bigcup \mathcal{S}(P_{\boldsymbol{\theta}_G^*})$ indicates that $D_{\boldsymbol{\theta}_D^*}$ may have no discrimination power. First, we make the following assumptions:

**Assumption 3.** *1. $D$ is a multilayer perceptron where each layer $l$ can be factorized into an affine transform and an element-wise activation function $f_l$. 2. Each activation function $f_l \in C^0$; furthermore, $f_l'$ has a finite number of discontinuities and $f_l'' \in C^0$[6]. 3. Input data to $D$ is continuous and its support $\mathcal{S}$ is compact in $\mathbb{R}^d$ with non-zero measure in each dimension and $d > 1$[7].*

Based on Assumption 3, we have the following proposition:

**Proposition 2.** *If $\forall \boldsymbol{x} \in \mathcal{S}$, $D(\boldsymbol{x}) = \boldsymbol{c}$, where $\boldsymbol{c}$ is constant, then there always exists distortion $\delta\boldsymbol{x}$ such that $\boldsymbol{x} + \delta\boldsymbol{x} \notin \mathcal{S}$ and $D(\boldsymbol{x} + \delta\boldsymbol{x}) = \boldsymbol{c}$.*

---

[6]This include many commonly used activations like linear, sigmoid, tanh, ReLU and ELU.

[7]For distributions with semi-infinite or infinite support, we consider the effective or truncated support $\mathcal{S}_\epsilon(P) = \{x \in \mathcal{X} | P(x) \geq \epsilon\}$, where $\epsilon > 0$ is a small scalar. This is practical, e.g., univariate Gaussian has support in $(-\infty, +\infty)$ yet a sample five standard deviations away from the mean is unlikely to be valid.

*Proof.* Without loss of generality, we consider $D(\boldsymbol{x}) = \boldsymbol{W}_2 h(\boldsymbol{x}) + \boldsymbol{b}_2$ and $h(\boldsymbol{x}) = f(\boldsymbol{W}_1 \boldsymbol{x} + \boldsymbol{b}_1)$, where $\boldsymbol{W}_1 \in \mathbb{R}^{d_h \times d}$, $\boldsymbol{W}_2$, $\boldsymbol{b}_1$, $\boldsymbol{b}_2$ are model weights and biases, $f$ is an activation function satisfying Assumption 3. For $\boldsymbol{x} \in \mathcal{S}$, since $D(\boldsymbol{x}) = c$, we have $h(\boldsymbol{x}) \in \text{null}(\boldsymbol{W}_2)$. Furthermore:

(a) If $\text{rank}(\boldsymbol{W}_1) < d$, for any $\delta \boldsymbol{x} \in \text{null}(\boldsymbol{W}_1)$, $h(\boldsymbol{x} + \delta \boldsymbol{x}) \in \text{null}(\boldsymbol{W}_2)$.

(b) If $\text{rank}(\boldsymbol{W}_1) = d = d_h$, the problem $h(\boldsymbol{x} + \delta \boldsymbol{x}) = k \cdot h(\boldsymbol{x})$ has unique solution for any $k \in \mathbb{R}$ as long as $k \cdot h(\boldsymbol{x})$ is within the output range of $f$.

(c) If $\text{rank}(\boldsymbol{W}_1) = d < d_h$, let $\boldsymbol{U}$ and $\boldsymbol{V}$ be two basis matrices of $R^{d_h}$ such that $\boldsymbol{W}_1 \boldsymbol{x} = \boldsymbol{U} \begin{bmatrix} \hat{\boldsymbol{x}}^T & \boldsymbol{0}^T \end{bmatrix}^T$ and any vector in $\text{null}(\boldsymbol{W}_2)$ can be represented as $\boldsymbol{V} \begin{bmatrix} \boldsymbol{z}^T & \boldsymbol{0}^T \end{bmatrix}^T$, where $\hat{\boldsymbol{x}} \in \mathbb{R}^{d_h \times d}$, $\boldsymbol{z} \in \mathbb{R}^{d_h \times n}$ and $n$ is the nullity of $\boldsymbol{W}_2$. Let the projected support be $\hat{\mathcal{S}}$. Thus, $\forall \hat{\boldsymbol{x}} \in \hat{\mathcal{S}}$, there exists $\boldsymbol{z}$ such that $f(\boldsymbol{U} \begin{bmatrix} \hat{\boldsymbol{x}}^T & \boldsymbol{0}^T \end{bmatrix}^T + \boldsymbol{b}_1) = \boldsymbol{V} \begin{bmatrix} \boldsymbol{z}^T & \boldsymbol{z}_c^T \end{bmatrix}^T$ with $\boldsymbol{z}_c = \boldsymbol{0}$. Consider the Jacobian:

$$\boldsymbol{J} = \frac{\partial \begin{bmatrix} \boldsymbol{z}^T & \boldsymbol{z}_c^T \end{bmatrix}^T}{\partial \begin{bmatrix} \hat{\boldsymbol{x}}^T & \boldsymbol{0}^T \end{bmatrix}^T} = \boldsymbol{V}^{-1} \nabla \Sigma \boldsymbol{U} \tag{S6}$$

where $\nabla \Sigma = \text{diag}(\frac{\mathrm{d} f}{\mathrm{d} a_i})$ and $\boldsymbol{a} = [a_i]_{i=1}^{d_h}$ is the input to activation, or pre-activations. Since $\hat{\mathcal{S}}$ is continuous and compact, it has infinite number of boundary points $\{\hat{\boldsymbol{x}}_b\}$ for $d > 1$. Consider one boundary point $\hat{\boldsymbol{x}}_b$ and its normal line $\delta \hat{\boldsymbol{x}}_b$. Let $\epsilon > 0$ be a small scalar such that $\hat{\boldsymbol{x}}_b - \epsilon \delta \hat{\boldsymbol{x}}_b \in \hat{\mathcal{S}}$ and $\hat{\boldsymbol{x}}_b + \epsilon \delta \hat{\boldsymbol{x}}_b \notin \hat{\mathcal{S}}$.

- For linear activation, $\nabla \Sigma = \boldsymbol{I}$ and $\boldsymbol{J}$ is constant. Then $\boldsymbol{z}_c$ remains $\boldsymbol{0}$ for $\hat{\boldsymbol{x}}_b + \epsilon \delta \hat{\boldsymbol{x}}_b$, i.e., there exists $\boldsymbol{z}$ such that $h(\hat{\boldsymbol{x}} + \epsilon \delta \hat{\boldsymbol{x}}) \in \text{null}(\boldsymbol{W}_2)$.

- For nonlinear activations, assume $f'$ has $N$ discontinuities. Since $\boldsymbol{U} \begin{bmatrix} \hat{\boldsymbol{x}}^T & \boldsymbol{0}^T \end{bmatrix}^T + \boldsymbol{b}_1 = \boldsymbol{c}$ has unique solution for any vector $\boldsymbol{c}$, the boundary points $\{\hat{\boldsymbol{x}}_b\}$ cannot yield pre-activations $\{\boldsymbol{a}_b\}$ that all lie on the discontinuities in any of the $d_h$ directions. Though we might need to sample $d_h^{N+1}$ points in the worst case to find an exception, there are infinite number of exceptions. Let $\hat{\boldsymbol{x}}_b$ be a sample where $\{\boldsymbol{a}_b\}$ does not lie on the discontinuities in any direction. Because $f''$ is continuous, $\boldsymbol{z}_c$ remains $\boldsymbol{0}$ for $\hat{\boldsymbol{x}}_b + \epsilon \delta \hat{\boldsymbol{x}}_b$, i.e., there exists $\boldsymbol{z}$ such that $h(\hat{\boldsymbol{x}} + \epsilon \delta \hat{\boldsymbol{x}}) \in \text{null}(\boldsymbol{W}_2)$.

In conclusion, we can always find $\delta \boldsymbol{x}$ such that $\boldsymbol{x} + \delta \boldsymbol{x} \notin \mathcal{S}$ and $D(\boldsymbol{x} + \delta \boldsymbol{x}) = \boldsymbol{c}$. □

Proposition 2 indicates that if $D_{\boldsymbol{\theta}_D^*}(\boldsymbol{x}) = \boldsymbol{0}$ for $\boldsymbol{x} \in \mathcal{S}(P_{\mathbf{X}}) \bigcup \mathcal{S}(P_{\boldsymbol{\theta}_G^*})$, $D_{\boldsymbol{\theta}_D^*}$ cannot discriminate against fake samples with distortions to the original data. In contrast, Assumption 2 and Lemma A.1 guarantee that, at equilibrium, the discriminator trained using MMD loss function is effective against such fake samples given a large number of i.i.d. test samples (Gretton et al. (2012)).

## B SUPPLEMENTARY METHODOLOGY

### B.1 REPRESENTATIVE LOSS FUNCTIONS IN LITERATURE

Several loss functions have been proposed to quantify the difference between real and generated sample scores, including: (assume linear activation is used at the last layer of $D$)

- The Minimax loss (Goodfellow et al. (2014)): $L_D = \mathbb{E}_{P_{\mathbf{X}}}[\text{Softplus}(-D(\boldsymbol{x}))] + \mathbb{E}_{P_{\mathbf{z}}}[\text{Softplus}(D(G(\boldsymbol{z})))]$ and $L_G = -L_D$, which can be derived from the Jensen–Shannon (JS) divergence between $P_{\mathbf{X}}$ and the model distribution $P_G$.

- The non-saturating loss (Goodfellow et al. (2014)), which is a variant of the minimax loss with the same $L_D$ and $L_G = \mathbb{E}_{P_{\mathbf{z}}}[\text{Softplus}(-D(G(\boldsymbol{z})))]$.

- The Hinge loss (Tran et al. (2017)): $L_D = \mathbb{E}_{P_{\mathbf{X}}}[\text{ReLU}(1 - D(\boldsymbol{x}))] + \mathbb{E}_{P_{\mathbf{z}}}[\text{ReLU}(1 + D(G(\boldsymbol{z})))]$, $L_G = \mathbb{E}_{P_{\mathbf{z}}}[-D(G(\boldsymbol{z}))]$, which is notably known for usage in support vector machines and is related to the total variation (TV) distance (Nguyen et al. (2009)).

- The Wasserstein loss (Arjovsky et al. (2017); Gulrajani et al. (2017)), which is derived from the Wasserstein distance between $P_{\mathbf{X}}$ and $P_G$: $L_G = -\mathbb{E}_{P_{\mathbf{z}}}[D(G(\mathbf{z}))]$, $L_D = \mathbb{E}_{P_{\mathbf{z}}}[D(G(\mathbf{z}))] - \mathbb{E}_{P_{\mathbf{X}}}[D(\mathbf{x})]$, where $D$ is subject to some Lipschitz constraint.

- The maximum mean discrepancy (MMD) (Li et al. (2017a); Bińkowski et al. (2018)), as described in Section 2.

## B.2 NETWORK ARCHITECTURE

For unsupervised image generation tasks on CIFAR-10 and STL-10 datasets, the DCGAN architecture from Miyato et al. (2018) was used. For CelebA and LSUN bedroom datasets, we added more layers to the generator and discriminator accordingly. See Table S1 and S2 for details.

Table S1: DCGAN models for image generation on CIFAR-10 ($h = w = 4$, $H = W = 32$) and STL-10 ($h = w = 6$, $H = W = 48$) datasets. For non-saturating loss and hinge loss, $s = 1$; for MMD-rand, MMD-rbf, MMD-rq, $s = 16$.

(a) Generator

| $\mathbf{z} \in \mathbb{R}^{128} \sim \mathcal{N}(\mathbf{0}, \mathbf{I})$ |
| --- |
| $128 \rightarrow h \times w \times 512$, dense, linear |
| $4 \times 4$, stride 2 deconv, 256, BN, ReLU |
| $4 \times 4$, stride 2 deconv, 128, BN, ReLU |
| $4 \times 4$, stride 2 deconv, 64, BN, ReLU |
| $3 \times 3$, stride 1 conv, 3, Tanh |

(b) Discriminator

| RGB image $\mathbf{x} \in [-1, 1]^{H \times W \times 3}$ |
| --- |
| $3 \times 3$, stride 1 conv, 64, LReLU |
| $4 \times 4$, stride 2 conv, 128, LReLU
$3 \times 3$, stride 1 conv, 128, LReLU |
| $4 \times 4$, stride 2 conv, 256, LReLU
$3 \times 3$, stride 1 conv, 256, LReLU |
| $4 \times 4$, stride 2 conv, 512, LReLU
$3 \times 3$, stride 1 conv, 512, LReLU |
| $h \times w \times 512 \rightarrow s$, dense, linear |

Table S2: DCGAN models for image generation on CelebA and LSUN-bedroom datasets. For non-saturating loss and hinge loss, $s = 1$; for MMD-rand, MMD-rbf, MMD-rq, $s = 16$.

(a) Generator

| $\mathbf{z} \in \mathbb{R}^{128} \sim \mathcal{N}(\mathbf{0}, \mathbf{I})$ |
| --- |
| $128 \rightarrow 4 \times 4 \times 1024$, dense, linear |
| $4 \times 4$, stride 2 deconv, 512, BN, ReLU |
| $4 \times 4$, stride 2 deconv, 256, BN, ReLU |
| $4 \times 4$, stride 2 deconv, 128, BN, ReLU |
| $4 \times 4$, stride 2 deconv, 64, BN, ReLU |
| $3 \times 3$, stride 1 conv, 3, Tanh |

(b) Discriminator

| RGB image $\mathbf{x} \in [-1, 1]^{64 \times 64 \times 3}$ |
| --- |
| $3 \times 3$, stride 1 conv, 64, LReLU |
| $4 \times 4$, stride 2 conv, 128, LReLU
$3 \times 3$, stride 1 conv, 128, LReLU |
| $4 \times 4$, stride 2 conv, 256, LReLU
$3 \times 3$, stride 1 conv, 256, LReLU |
| $4 \times 4$, stride 2 conv, 512, LReLU
$3 \times 3$, stride 1 conv, 512, LReLU |
| $4 \times 4$, stride 2 conv, 1024, LReLU
$3 \times 3$, stride 1 conv, 1024, LReLU |
| $4 \times 4 \times 512 \rightarrow s$, dense, linear |

## C  POWER ITERATION FOR CONVOLUTION OPERATION

This section introduces the power iteration for convolution operation (PICO) method to estimate the spectral norm of a convolution kernel, and compare PICO with the power iteration for matrix (PIM) method used in Miyato et al. (2018).

### C.1  METHOD FORMATION

For a weight matrix $\boldsymbol{W}$, the spectral norm is defined as $\sigma(\boldsymbol{W}) = \max_{\|\boldsymbol{v}\|_2 \leq 1} \|\boldsymbol{W}\boldsymbol{v}\|_2$. The PIM is used to estimate $\sigma(\boldsymbol{W})$ (Miyato et al. (2018)), which iterates between two steps:

1. Update $\boldsymbol{u} = \boldsymbol{W}\boldsymbol{v} / \|\boldsymbol{W}\boldsymbol{v}\|_2$;
2. Update $\boldsymbol{v} = \boldsymbol{W}^T\boldsymbol{u} / \|\boldsymbol{W}^T\boldsymbol{u}\|_2$.

The convolutional kernel $\boldsymbol{W}_c$ is a tensor of shape $h \times w \times c_{in} \times c_{out}$ with $h, w$ the receptive field size and $c_{in}, c_{out}$ the number of input/output channels. To estimate $\sigma(\boldsymbol{W}_c)$, Miyato et al. (2018) reshaped it into a matrix $\boldsymbol{W}_{rs}$ of shape $(hwc_{in}) \times c_{out}$ and estimated $\sigma(\boldsymbol{W}_{rs})$.

We propose a simple method to calculate $\boldsymbol{W}_c$ directly based on the fact that convolution operation is linear. For any linear map $T : \mathbb{R}^m \to \mathbb{R}^n$, there exists matrix $\boldsymbol{W}_L \in \mathbb{R}^{n \times m}$ such that $\boldsymbol{y} = T(\boldsymbol{x})$ can be represented as $\boldsymbol{y} = \boldsymbol{W}_L\boldsymbol{x}$. Thus, we may simply substitute $\boldsymbol{W}_L = \frac{\partial \boldsymbol{y}}{\partial \boldsymbol{x}}$ in the PIM method to estimate the spectral norm of any linear operation. In the case of convolution operation $*$, there exist doubly block circulant matrix $\boldsymbol{W}_{dbc}$ such that $\boldsymbol{u} = \boldsymbol{W}_c * \boldsymbol{v} = \boldsymbol{W}_{dbc}\boldsymbol{v}$. Consider $\boldsymbol{v}' = \boldsymbol{W}_{dbc}^T\boldsymbol{u} = [\frac{\partial \boldsymbol{u}}{\partial \boldsymbol{v}}]^T\boldsymbol{u}$ which is essentially the transpose convolution of $\boldsymbol{W}_c$ on $\boldsymbol{u}$ (Dumoulin & Visin (2016)). Thus, similar to PIM, PICO iterates between the following two steps:

1. Update $\boldsymbol{u} = \boldsymbol{W}_c * \boldsymbol{v} / \|\boldsymbol{W}_c * \boldsymbol{v}\|_2$;
2. Do transpose convolution of $\boldsymbol{W}_c$ on $\boldsymbol{u}$ to get $\hat{\boldsymbol{v}}$; update $\boldsymbol{v} = \hat{\boldsymbol{v}} / \|\hat{\boldsymbol{v}}\|_2$.

Similar approaches have been proposed in Tsuzuku et al. (2018) and Virmaux & Scaman (2018) from different angles, which we were not aware during this study. In addition, Sedghi et al. (2019) proposes to compute the exact singular values of convolution kernels using FFT and SVD. In spectral normalization, only the first singular value is concerned, making the power iteration methods PIM and PICO more efficient than FFT and thus preferred in our study. However, we believe the exact method FFT+SVD (Sedghi et al. (2019)) may eventually inspire more rigorous regularization methods for GAN.

The proposed PICO method estimates the real spectral norm of a convolution kernel at each layer, thus enforces an upper bound on the Lipschitz constant of the discriminator $D$. Denote the upper bound as LIP$_{\text{PICO}}$. In this study, Leaky ReLU (LReLU) was used at each layer of $D$, thus LIP$_{\text{PICO}} \approx 1$ (Virmaux & Scaman (2018)). In practice, however, PICO would often cause the norm of the signal passing through $D$ to decrease to zero, because at each layer,

- the signal hardly coincides with the first singular-vector of the convolution kernel; and
- the activation function LReLU often reduces the norm of the signal.

Consequently, the discriminator outputs tend to be similar for all the inputs. To compensate the loss of norm at each layer, the signal is multiplied by a constant $C$ after each spectral normalization. This essentially enlarges LIP$_{\text{PICO}}$ by $C^K$ where $K$ is the number of layers in the DCGAN discriminator. For all experiments in Section 5, we fixed $C = \frac{1}{0.55} \approx 1.82$ as all loss functions performed relatively well empirically. In Appendix Section C.3, we tested the effects of coefficient $C^K$ on the performance of several loss functions.

### C.2  COMPARISON TO PIM

PIM (Miyato et al. (2018)) also enforces an upper bound LIP$_{\text{PIM}}$ on the Lipschitz constant of the discriminator $D$. Consider a convolution kernel $\boldsymbol{W}_c$ with receptive field size $h \times w$ and stride $s$. Let $\sigma_{\text{PICO}}$ and $\sigma_{\text{PIM}}$ be the spectral norm estimated by PICO and PIM respectively. We empirically

Table S3: Fréchet Inception distance (FID) on image generation tasks using spectral normalization with two power iteration methods PICO and PIM

| Methods | $C^K$ in PICO | CIFAR-10 | | STL-10 | | CelebA | | LSUN-bedrom | |
|---|---|---|---|---|---|---|---|---|---|
| | | PIM | PICO | PIM | PICO | PIM | PICO | PIM | PICO |
| Hinge | 128 | 23.60 | 22.89 | 47.10 | 47.24 | 10.02 | 9.08 | 27.38 | 17.20 |
| MMD-rbf | 128 | 26.56 | 26.50 | 53.17 | 54.23 | 13.06 | 12.81 | | |
| MMD-rep | 64 | 19.98 | 17.00 | 40.40 | 37.15 | 8.51 | 6.81 | 74.03 | 16.01 |
| MMD-rep-b | 64 | 18.24 | 16.65 | 39.78 | 37.31 | 7.09 | 6.42 | 20.12 | 11.22 |

found[8] that $\sigma_{\text{PIM}}^{-1}$ varies in the range $[\sigma_{\text{PICO}}^{-1}, \frac{\sqrt{hw}}{s}\sigma_{\text{PICO}}^{-1}]$, depending on the kernel $\boldsymbol{W}_c$. For a typical kernel of size $3 \times 3$ and stride 1, $\sigma_{\text{PIM}}^{-1}$ may vary from $\sigma_{\text{PICO}}^{-1}$ to $3\sigma_{\text{PICO}}^{-1}$. Thus, $\text{LIP}_{\text{PIM}}$ is indefinite and may vary during training. In deep convolutional networks, PIM could potentially result in a very loose constraint on the Lipschitz constant of the network. In Appendix Section C.3, we experimentally compare the performance of PICO and PIM with several loss functions.

## C.3 EXPERIMENTS

In this section, we empirically evaluate the effects of coefficient $C^K$ on the performance of PICO and compare PICO against PIM using several loss functions.

**Experiment setup:** We used a similar setup as Section 5.1 with the following adjustments. Four loss functions were tested: hinge, MMD-rbf, MMD-rep and MMD-rep-b. Either PICO or PIM was used at each layer of the discriminator. For PICO, five coefficients $C^K$ were tested: 16, 32, 64, 128 and 256 (note this is the overall coefficient for $K$ layers; $K = 8$ for CIFAR-10 and STL-10; $K = 10$ for CelebA and LSUN-bedroom; see Appendix B.2). FID was used to evaluate the performance of each combination of loss function and power iteration method, e.g., hinge + PICO with $C^K = 16$.

**Results:** For each combination of loss function and power iteration method, the distribution of FID scores over 16 learning rate combinations is shown in Fig. S2. We separated well-performed learning rate combinations from diverged or poorly-performed ones using a threshold $\tau$ as the diverged cases often had non-meaningful FID scores. The boxplot shows the distribution of FID scores for good-performed cases while the number of diverged or poorly-performed cases was shown above each box if it is non-zero.

Fig. S2 shows that:

1) When PICO was used, the hinge, MMD-rbf and MMD-rep methods were sensitive to the choices of $C^K$ while MMD-rep-b was robust. For hinge and MMD-rbf, higher $C^K$ may result in better FID scores and less diverged cases over 16 learning rate combinations. For MMD-rep, higher $C^K$ may cause more diverged cases; however, the best FID scores were often achieved with $C^K = 64$ or 128.

2) For CIFAR-10, STL-10 and CelebA datasets, PIM performed comparable to PICO with $C^K = 128$ or 256 on four loss functions. For LSUN bedroom dataset, it is likely that the performance of PIM corresponded to that of PICO with $C^K > 256$. This implies that PIM may result in a relatively loose Lipschitz constraint on deep convolutional networks.

3) MMD-rep-b performed generally better than hinge and MMD-rbf with tested power iteration methods and hyper-parameter configurations. Using PICO, MMD-rep also achieved generally better FID scores than hinge and MMD-rbf. This implies that, given a limited computational budget, the proposed repulsive loss may be a better choice than the hinge and MMD loss for the discriminator.

---

[8]This was obtained by optimizing $\sigma_{\text{PICO}}/\sigma_{\text{PIM}}$ w.r.t. a variety of randomly initialized kernel $\boldsymbol{W}_c$. Both gradient descent and Adam methods were tested with a small learning rate $1e^{-5}$ so that the error of spectral norm estimation may be ignored at each iteration.

Table S4: Inception score (IS) and Fréchet Inception distance (FID) on CIFAR-10 dataset using gradient penalty and different loss functions

| Methods[1] | IS | FID |
|---|---|---|
| Real data | 11.31 | 2.09 |
| SMMDGAN[2] | 7.0 | 31.5 |
| SN-SMMDGAN[2] | **7.3** | 25.0 |
| MMD-rep-gp | 7.26 | **23.01** |

[1] All methods used the same DCGAN architecture.
[2] Results from Arbel et al. (2018) Table 1.

Table S5: Inception score (IS), Fréchet Inception distance (FID) on CIFAR-10 dataset using MMD-rep and different dimensions of discriminator outputs

| Methods | $C^K$ | IS | FID |
|---|---|---|---|
| Real data | | 11.31 | 2.09 |
| MMD-rep-1 | 64 | 7.43 | 22.43 |
| MMD-rep-4 | 64 | 7.81 | 17.87 |
| MMD-rep-16 | 32 | 8.20 | 16.99 |
| MMD-rep-64 | 32 | 8.08 | 15.65 |
| MMD-rep-256 | 32 | 7.96 | 16.61 |

Table S3 shows the best FID scores obtained by PICO and PIM where $C^K$ was fixed at $128$ for hinge and MMD-rbf, and $64$ for MMD-rep and MMD-rep-b. For hinge and MMD-rbf, PICO performed significantly better than PIM on the LSUN-bedroom dataset and comparably on the rest datasets. For MMD-rep and MMD-rep-b, PICO achieved consistently better FID scores than PIM.

However, compared to PIM, PICO has a higher computational cost which roughly equals the additional cost incurred by increasing the batch size by two (Tsuzuku et al. (2018)). This may be problematic when a small batch has to be used due to memory constraints, e.g., when handling high resolution images on a single GPU. Thus, we recommend using PICO when the computational cost is less of a concern.

## D  Supplementary Experiments

### D.1  Lipschitz Constraint via Gradient Penalty

Gradient penalty has been widely used to impose the Lipschitz constraint on the discriminator arguably since Wasserstein GAN (Gulrajani et al. (2017)). This section explores whether the proposed repulsive loss can be applied with gradient penalty.

Several gradient penalty methods have been proposed for MMD-GAN. Bińkowski et al. (2018) penalized the gradient norm of witness function $f_w(\boldsymbol{z}) = \mathbb{E}_{P_\mathbf{X}}[k_D(\boldsymbol{z}, \boldsymbol{x})] - \mathbb{E}_{P_G}[k_D(\boldsymbol{z}, \boldsymbol{y})]$ w.r.t. the interpolated sample $\boldsymbol{z} = u\boldsymbol{x} + (1-u)\boldsymbol{y}$ to one, where $u \sim \mathcal{U}(0, 1)$[9]. More recently, Arbel et al. (2018) proposed to impose the Lipschitz constraint on the mapping $\phi \circ D$ directly and derived the Scaled MMD (SMMD) as $SM_k(P, Q) = \sigma_{\mu,k,\lambda} M_k(P, Q)$, where the scale $\sigma_{\mu,k,\lambda}$ incorporates gradient and smooth penalties. Using the Gaussian kernel and measure $\mu = P_\mathbf{X}$ leads to the discriminator loss:

$$L_D^{\text{SMMD}} = \frac{L_D^{\text{att}}}{1 + \lambda \mathbb{E}_{P_\mathbf{X}}[\|\nabla D(\boldsymbol{x})\|_F^2]} \tag{S7}$$

We apply the same formation of gradient penalty to the repulsive loss:

$$L_D^{\text{rep-gp}} = \frac{L_D^{\text{rep}} - 1}{1 + \lambda \mathbb{E}_{P_\mathbf{X}}[\|\nabla D(\boldsymbol{x})\|_F^2]} \tag{S8}$$

where the numerator $L_D^{\text{rep}} - 1 \leq 0$ so that the discriminator will always attempt to minimize both $L_D^{\text{rep}}$ and the Frobenius norm of gradients $\nabla D(\boldsymbol{x})$ w.r.t. real samples. Meanwhile, the generator is trained using the MMD loss $L_G^{\text{mmd}}$ (Eq. 2).

**Experiment setup:** The gradient-penalized repulsive loss $L_D^{\text{rep-gp}}$ (Eq. S8, referred to as MMD-rep-gp) was evaluated on the CIFAR-10 dataset. We found $\lambda = 10$ in Arbel et al. (2018) too restrictive

[9]Empirically, we found this gradient penalty did not work with the repulsive loss. The reason may be the attractive loss $L_D^{\text{att}}$ (Eq. 3) is symmetric in the sense that swapping $P_\mathbf{X}$ and $P_G$ results in the same loss; while the repulsive loss is asymmetric and naturally results in varying gradient norms in data space.

and used $\lambda = 0.1$ instead. Same as Arbel et al. (2018), the output dimension of discriminator was set to one. Since we entrusted the Lipschitz constraint to the gradient penalty, spectral normalization was not used. The rest experiment setup can be found in Section 5.1.

**Results:** Table S4 shows that the proposed repulsive loss can be used with gradient penalty to achieve reasonable results on CIFAR-10 dataset. For comparison, we cited the Inception score and FID for Scaled MMD-GAN (SMMDGAN) and Scaled MMD-GAN with spectral normalization (SN-SMMDGAN) from Arbel et al. (2018). Note that SMMDGAN and SN-SMMDGAN used the same DCGAN architecture as MMD-rep-gp, but were trained for 150k generator updates and 750k discriminator updates, much more than that of MMD-rep-gp (100k for both $G$ and $D$). Thus, the repulsive loss significantly improved over the attractive MMD loss for discriminator.

## D.2    OUTPUT DIMENSION OF DISCRIMINATOR

In this section, we investigate the impact of the output dimension of discriminator on the performance of repulsive loss.

**Experiment setup:** We used a similar setup as Section 5.1 with the following adjustments. The repulsive loss was tested on the CIFAR-10 dataset with a variety of discriminator output dimensions: $d \in \{1, 4, 16, 64, 256\}$. Spectral normalization was applied to discriminator with the proposed PICO method (see Appendix C) and the coefficients $C^K$ selected from $\{16, 32, 64, 128, 256\}$.

**Results:** Table S5 shows that using more than one output neuron in the discriminator $D$ significantly improved the performance of repulsive loss over the one-neuron case on CIFAR-10 dataset. The reason may be that using insufficient output neurons makes it harder for the discriminator to learn an injective and discriminative representation of the data (see Fig. 4b). However, the performance gain diminished when more neurons were used, perhaps because it becomes easier for $D$ to surpass the generator $G$ and trap it around saddle solutions. The computation cost also slightly increased due to more output neurons.

## D.3    SAMPLES OF UNSUPERVISED IMAGE GENERATION

Generated samples on CelebA dataset are given in Fig. S3 and LSUN bedrooms in Fig. S4.

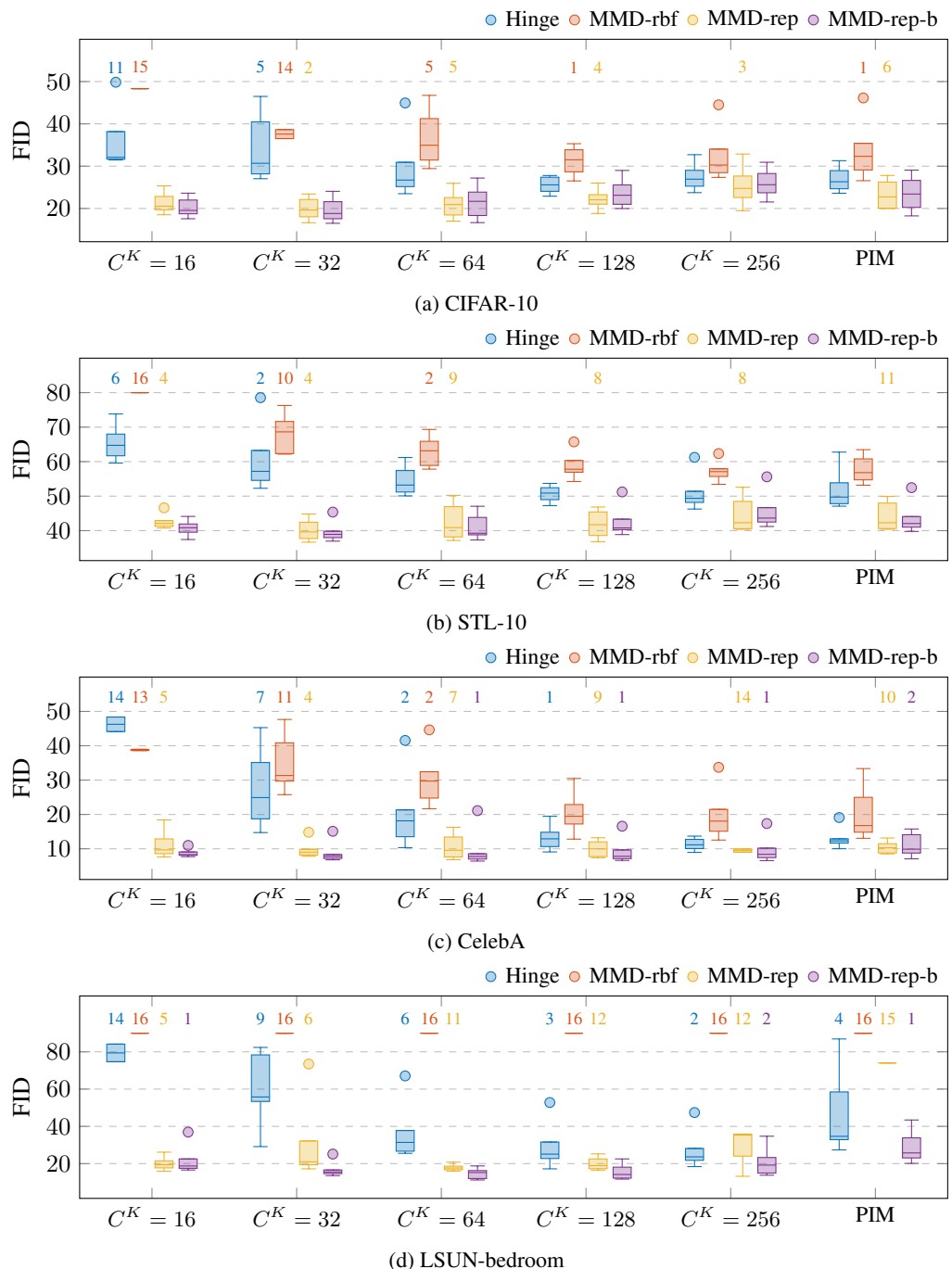

Figure S2: Boxplot of the FID scores for 16 learning rate combinations on four datasets: (a) CIFAR-10, (b) STL-10, (c) CelebA, (d) LSUN-bedroom, using four loss functions, Hinge, MMD-rbf, MMD-rep and MMD-rmb. Spectral normalization was applied to discriminator with two power iteration methods: PICO and PIM. For PICO, five coefficients $C^K$ were tested: 16, 32, 64, 128, and 256. A learning rate combination was considered diverged or poorly-performed if the FID score exceeded a threshold $\tau$, which is 50, 80, 50, 90 for CIFAR-10, STL-10, CelebA and LSUN-bedroom respectively. The box quartiles were plotted based on the cases with FID $< \tau$ while the number of diverged or poorly-performed cases (out of 16 learning rate combinations) was shown above each box if it is non-zero. We introduced $\tau$ because the diverged cases often had arbitrarily large and non-meaningful FID scores.

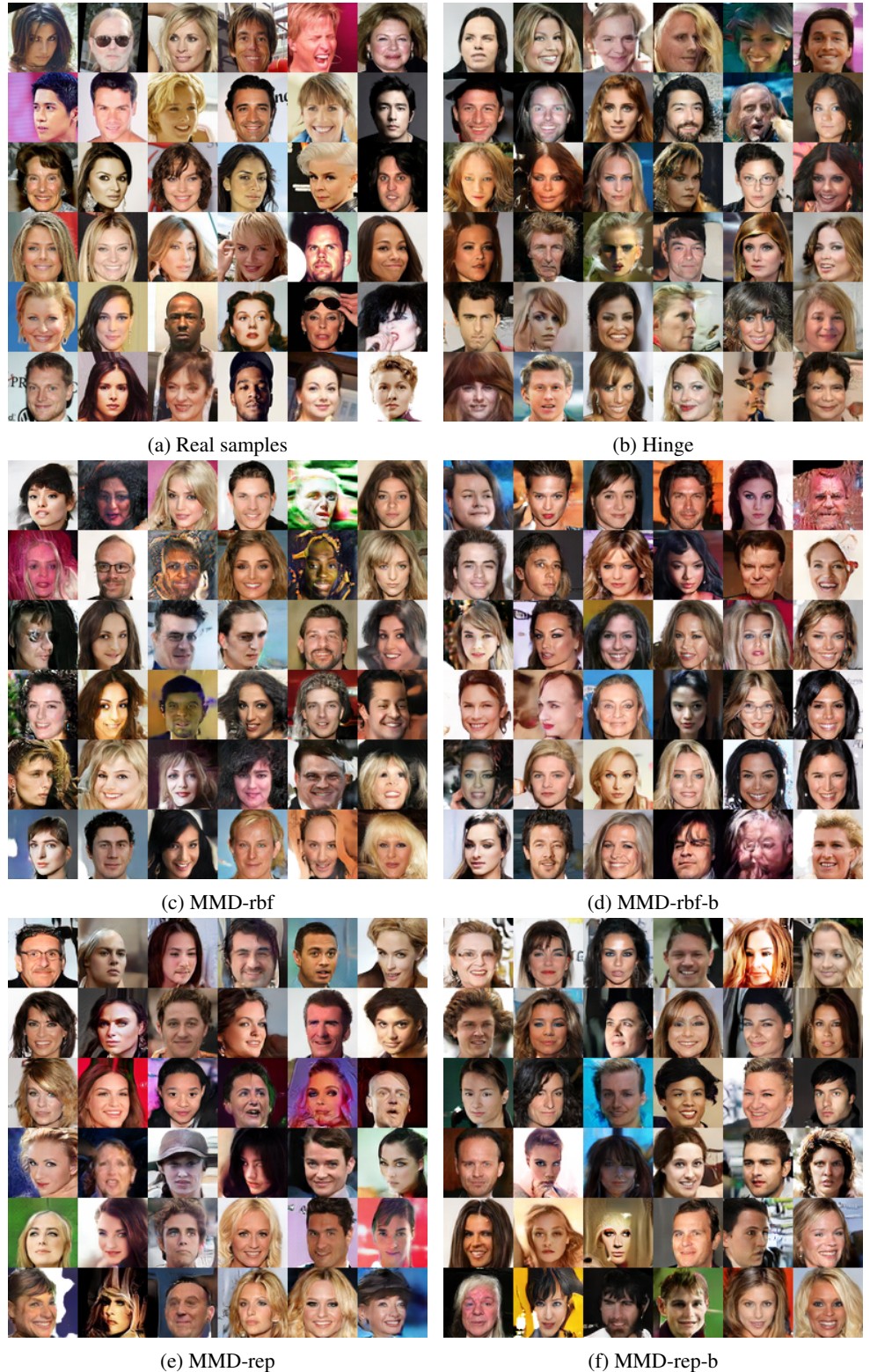

Figure S3: Image generation using different loss functions on $64 \times 64$ CelebA dataset.

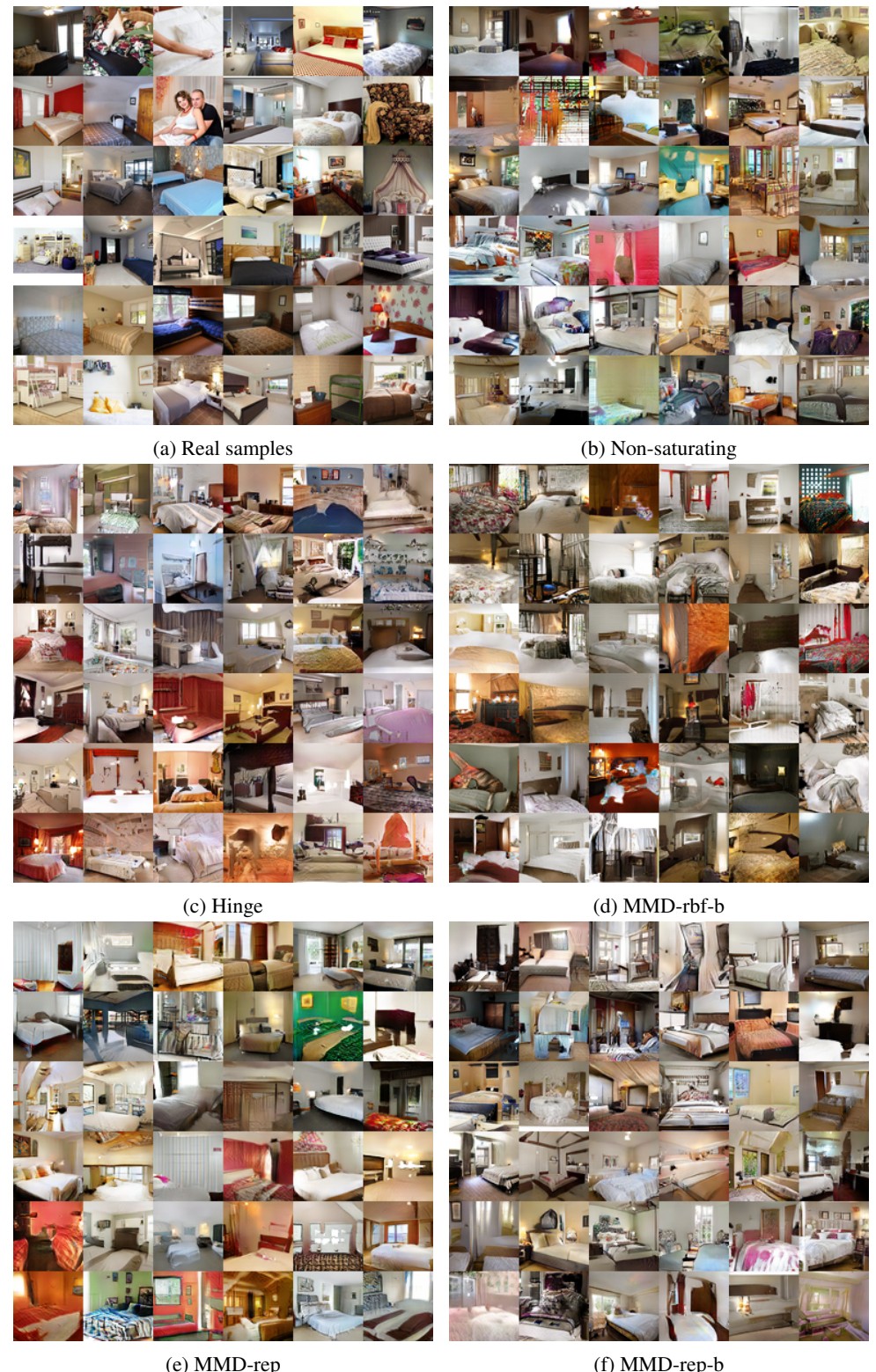

(a) Real samples

(b) Non-saturating

(c) Hinge

(d) MMD-rbf-b

(e) MMD-rep

(f) MMD-rep-b

Figure S4: Image generation using different loss functions on $64 \times 64$ LSUN bedroom dataset.

