# OpenReview forum: "Improving MMD-GAN Training with Repulsive Loss Function"
_ICLR.cc/2019/Conference_

### Official Review · AnonReviewer2 · 2018-11-01
**Interesting idea but more evidence to show the significance of the work would be appreciated.**

**Rating:** 7
**Confidence:** 5

**Review:**

The paper proposes a new discriminator loss for MMDGAN which encourages repulsion between points from the target distribution. The discriminator can then learn finer details of the target distribution unlike previous versions of MMDGAN. The paper also proposes an alternative to the RBF kernel to stabilize training and use spectral normalization to regularize the discriminator. The paper is clear and well written overall and the experiments show that the proposed method leads to improvements. The proposed idea is promising and a better theoretical understanding would make this work more significant. Indeed, it seems that MMD-rep can lead to instabilities during training while this is not the case for MMD-rep as shown in Appendix A. It would be good to better understand under which conditions MMD-rep leads to stable training. Figure 3 suggests that lambda should not be too big, but more theoretical evidence would be appreciated.
Regarding the experiments:
- The proposed repulsive loss seems to improve over the classical attractive loss according to table 1, however, some ablation studies might be needed: how much improvement is attributed to the use of SN alone? The Hinge loss uses 1 output dimension for the critic and still leads to good results, while MMD variants use 16 output dimensions. Have you tried to compare the methods using the same dimension?
-The generalized spectral normalization proposed in this work seems to depend on the dimensionality of the input which can be problematic for high dimensional inputs. On the other hand, Myato’s algorithm only depends on the dimensions of the filter. Moreover, I would expect the two spectral norms to be mathematically related [1]. It is unclear what advantages the proposed algorithm for computing SN has.
- Regarding the choice of the kernel, it doesn’t seem that the choice defined in eq 6 and 7 defines a positive semi-definite kernel because of the truncation and the fact that it depends on whether the input comes from the true or the fake distribution. In that case, the mmd loss loses all its interpretation as a distance. Besides, the issue of saturation of the Gaussian kernel was already addressed in a more general case in [2]. Is there any reason to think the proposed kernel has any particular advantage?

Revision:

After reading the author's response, I think most of the points were well addressed and that the repulsive loss has interesting properties that should be further investigated. Also, the authors show experimentally the benefit of using PICO ver PIM which is also an interesting finding.
I'm less convinced by the bounded RBF kernel, which seems a little hacky although it works well in practice. I think the saturation issues with RBF kernel is mainly due to discontinuity under the weak topology of the optimized MMD [2] and can be fixed by controlling the Lipschitz constant of the critic.
Overall I feel that this paper has two interesting contributions (Repulsive loss + highlighting the difference between PICO and PIM) and I would recommend acceptance.






[1]: Sedghi, Hanie, Vineet Gupta, and Philip M. Long. “The Singular Values of Convolutional Layers.” CoRR
[2]: M. Arbel, D. J. Sutherland, M. Binkowski, and A. Gretton. On gradient regularizers for MMD GANs.

---

> ### Author Response · Authors · 2018-11-26
> **Response to Reviewer 2 comments**
>
> Thank you for your constructive comments. Below we would try to address your concerns.
>
> Q1: It seems that MMD-rep can lead to instabilities during training while this is not the case for MMD-rep as shown in Appendix A. What are the conditions MMD-rep leads to stable training?
> A1: We would like to point out that the training stability is different from the local stability considered in Appendix A.
>
> Appendix A demonstrates the local stability of MMD-rep. That is, if MMD-rep is initialized sufficiently close to an equilibrium and trained by gradient descent, it will converge to the equilibrium. In contrast, Wasserstein GAN does not have this property [1].
>
> In practice, training stability often refers to the ability of model converging to a desired state measured by some criterion. The repulsive loss may result in unstable training, due to factors including initialization (see Appendix A.2 and Fig. S1), learning rate (see Fig. 3) and Lipschitz constraints imposed by the proposed spectral normalization method (see Appendix C3. and Fig. S2).
>
> In diverged cases, we often observed that the discriminator outputs caused the Gaussian kernel to saturate. To alleviate this issue, we proposed the bounded Gaussian kernel. Fig. 3 and Appendix Fig. S2 show that the bounded kernel stabilized MMD-rep training in many cases.
>
> Q2: Figure 3 suggests that lambda should not be too big, but more theoretical evidence would be appreciated.
> A2: We suspect the reason is larger lambda leads to more focus on repulsing real sample scores. Consider lambda>>1, the model would simply 1) expand real sample scores, 2) pull generated sample scores to real samples’, and 3) ignore the attraction on generated sample scores. This process is divergent. We included this in Section 5.2 Paragraph 2 of the revised manuscript.
>
> Q3: How much improvement in Table 1 is attributed to spectral normalization? For hinge loss, the discriminator uses 1 output neuron; for repulsive loss, it is 16. How about repulsive loss with 1 output neuron?
> A3: We would like to point out that spectral normalization was used for every loss function in Table 1. In addition, Appendix Fig. S2 shows the results for other spectral normalization configurations. Given almost identical experiment setups, we attribute the improvement of MMD-rep and MMD-rep-b over MMD-rbf and MMD-rbf-b to the proposed repulsive loss. We clarified this in Section 5.2 Paragraph 1 of the revised manuscript.
>
> In Appendix D.2, we evaluated MMD-rep with various discriminator output dimensions: 1, 4, 16, 64, 256 on CIFAR-10 dataset; and found that the performance can be significantly improved using more than one output neuron. Additionally, MMD-rep with 1 discriminator output neuron was slightly better than the hinge loss.
>
> Q4: Comparison between the proposed generalized power iteration method and the one in [2]
> A4: In Appendix C.2 and C.3 of the revised manuscript, we compared the proposed power iteration for convolution kernel (PICO) against the method for matrix (PIM) used in [2]. In summary,
> 1) the spectral norm estimated by PIM may vary in a range related to the spectral norm by PICO;
> 2) PIM impose an indefinite and often loose upper bound on the Lipschitz constant of discriminator;
> 3) PICO performed better than PIM on cases using repulsive loss.
> We admit the PICO has higher computational cost than PIM, esp. when a small batch size has to be used in training. We recommend using PICO when the computational cost is less of a concern.
>
> Q5: Using the bounded RBF kernel, the MMD loss cannot be interpreted as a distance.
> A5: We would like to point out that the bounded RBF kernel is only used in the discriminator loss. The generator always attempts to minimize the MMD loss with a characteristic kernel. We highlighted this in Section 4.1 of the revised manuscript.
>
> Q6: The issue of saturation of the Gaussian kernel was already addressed in a more general case in [3]. Is there any advantage of the proposed kernel?
> A6: The gradient penalty from Scaled MMD of [3] is designed to impose a Lipschitz constraint on the discriminator w.r.t. real samples. We argue the method may have only partially addressed the saturation issue, as the following two scenarios may cause saturation: 1) the real sample scores may be very similar as encouraged by both the MMD loss and gradient penalty; 2) the generated sample scores may be very distinct or similar as the gradient penalty has no effects w.r.t. the generated samples.
>
> The proposed bounded kernel is designed to address the saturation issue, with the advantage of low computational cost. However, it does not impose Lipschitz constraints and may need to be used with methods like the gradient penalty from [3].
>
> -------------------------------------------------
> [1] Gradient descent GAN optimization is locally stable. NIPS, 2017.
> [2] Spectral Normalization for Generative Adversarial Networks. ICLR, 2018.
> [3] On Gradient Regularizers for MMD GANs. NIPS, 2018.

---

### Official Review · AnonReviewer3 · 2018-11-04

**Rating:** 7
**Confidence:** 5

**Review:**

This paper proposed two techniques to improve MMD GANs: 1) a repulsive loss for MMD loss optimization; 2) a bounded Gaussian RBF kernel instead of original Gaussian kernel. The experimental results on several benchmark shown the effectiveness of the two proposals. The paper is well written and the idea is somehow novel.

Despite the above strong points, here are some of my concerns:
1.The two proposed solutions seem separated. Do the authors have any clue that they can achieve more improvement when combined together, and why?

2. They are limited to the cases with spectral normalization. Is there any way both trick can be extended to other tricks (like WGAN loss case or GP).

3. Few missed references in this area:
a. On gradient regularizers for MMD GANs
b. Regularized Kernel and Neural Sobolev Descent: Dynamic MMD Transport

Revision: after reading rebuttal (as well as to other reviewers), I think they addressed my concerns. I would like to keep the original score.

---

> ### Author Response · Authors · 2018-11-26
> **Response to Reviewer 3 comments**
>
> We appreciate your valuable comments. We would try to address your concerns below.
>
> Q1: The proposed repulsive loss and bounded RBF kernel seem separated. Would they achieve better performance when combined and why?
> A1: In the revised manuscript, Fig. 3 and Appendix Fig. S2 show that the repulsive loss may result in unstable training, where we often observed that the discriminator outputs caused the Gaussian kernel to saturate. This issue motivated us to propose the bounded kernel. Table 1, Fig. 3 and Appendix Fig. S2 show that the repulsive loss combined with bounded kernel achieved comparable or better performance than the repulsive loss alone. Moreover, the bounded kernel managed to stabilize MMD-rep training under a variety of learning rate combinations and spectral normalization configurations.
>
> Q2: The experiments are limited to the cases with spectral normalization. Can both tricks be extended to other tricks (like WGAN loss or gradient penalty)?
> A2: We agree with the reviewer that it would be interesting to test the repulsive loss and bounded kernel in more general experiment setups, e.g., ResNet, gradient penalty, self-attention modules, supervised training, etc. In Appendix D.1 of revised manuscript, we show that the repulsive loss performed well using the gradient penalty from [1]. However, we are afraid to admit that a comprehensive study with other setups would require substantially more computational resources. We would continue our study to fill in this gap in the future.
>
> -------------------------------------------------
>  [1]: On Gradient Regularizers for MMD GANs. NIPS, 2018.

---

### Official Review · AnonReviewer1 · 2018-11-08

**Rating:** 6
**Confidence:** 2

**Review:**

OVERALL COMMENTS:

I haven't had much time to write this, so I'm giving a low confidence score and you should feel free to correct me.

I didn't think this paper was very clear.
I had trouble grasping what the contributions were supposed to be
and I had trouble judging the significance of the experiments.

That said, now that (I think) I understand what's going on,
the idea seems well motivated, the connection between the repulsion and the use of label information in other
GAN variants makes sense to me, and the statements you are making seem (as much as I had time to check them) correct.

This leaves the issue of scientific significance.
I feel like I need to understand what specifically contributed to the improvements in table 1 to evaluate significance.
First of all, it seems like there are a lot of other 'good-scoring' models left out of this table.
I understand that you make the claim that your improvement is orthogonal, but that seems like something that needs to
be tested empirically. You have orthogonal motivation but it might be that in practice your technique works for a reason
similar to the reason other techniques work. I would like to see more exploration of this.
Second, are the models below the line the only models using spectral norm? I can't tell.
Overall, it's hard for me to envision this work really seriously changing the course of research on GANs,
but that's perhaps too high a bar for poster acceptance.

For these reasons, I am giving a score of 6.

DETAILED COMMENTS ON TEXT:

> their performance heavily depends on the loss functions used in training.
This is not true, IMO. See [1]


> may discourage the learning of data structures
What does 'data structures' mean in this case?
It has another more common usage that makes this confusing.

> Several loss functions have been proposed
IMO this list doesn't belong in the main body of the text.
I would move it to an appendix.

> We assume linear activation is used at the last layer of D
I'm not sure what this means?
My best guess is just that you're saying there is no activation function applied to the logits.

> Arjovsky et al. (2017) showed that, if the supports of PX and PG do not overlap, there exists a perfect discriminator...
This doesn't affect your paper that much, but was this really something that needed to be shown?
If the discriminator has finite capacity it's not true in general and if it has infinite capacity its vacuous.


> We propose a generalized power iteration method...
Why do this when we can explicitly compute the singular values as in [2]?
Genuine question.

> MS-SSIM is not compatible with CIFAR-10 and STL-10 which have data from many classes;
Just compute the intra-class MS-SSIM as in [3].

> Higher IS and lower FID scores indicate better image quality
I'm a bit worried about using the FID to evaluate a model that's been trained w/ an MMD loss where
the discriminator is itself a neural network w/ roughly the same architecture as the pre-trained image classifier
used to compute the FID. What can you say about this?
Am I wrong to be worried?

> Table 1:
Which models use spectral norm?
My understanding is that this has a big influence on the scores.
This seems like a very important point.



REFERENCES:

[1] Are GANs Created Equal? A Large-Scale Study
[2] The Singular Values of Convolutional Layers
[3] Conditional Image Synthesis With Auxiliary Classifier GANs

---

> ### Author Response · Authors · 2018-11-08
> **Response to Reviewer 1 comments**
>
> Thank you for your precious comments. Below we would try to clarify our study and address your concerns.
>
> Q1. What specifically contributed to the improvements in Table 1. Other good-scoring models need to be tested empirically.
> A1: In this study, we focused on comparing the proposed repulsive loss function with other representative loss functions. The experiments in Table 1 were done in an almost identical setup: DCGAN + spectral normalization + Adam + 16 learning rate combinations + 100k iterations (see Section 5.1 Experiment Setup). That is, the methods in Table 1 differ mainly in the loss functions used. Thus, we attribute the improvements to our proposed repulsive loss. We highlighted this in Table 1 note 1 and Sec. 5.2 in the revised manuscript.
>
> The experiment setup was "almost identical" because, for MMD-related losses, the output layer of DCGAN has 16 neurons, while for logistic and hinge losses, it is one. In Appendix D.2 of the revised manuscript, we tested the discriminator with 1, 4, 16, 64, 256 output neurons and show that repulsive loss performed better when more than one output neuron was used.
>
> We agree that it would be interesting to test the repulsive loss in more general experiment setups, e.g., ResNet, gradient penalty, self-attention modules, supervised training, etc. In Appendix D.1, we show that the repulsive loss performed well using the gradient penalty from [1]. However, we are afraid to admit that a comprehensive study using other setups would require substantially more computational resources. We would try our best to fill in this gap in the future.
>
> Q2: Does GAN performance heavily depend on the loss functions used in training?
> A2: We agree that this is an overstatement and changed this to: "their performance may heavily depend on the loss functions, given a limited computational budget". [2], [3] and our study did find that different loss functions lead to quite different performances in practice with a limited computational budget.
>
> Q3: What does 'data structure' mean in that case that MMD may discourage learning of data structure?
> A3: In the revised manuscript, we have changed “data structure” to “differences among real data”. These differences, or fine details, separate the real samples. For example, in CIFAR-10 dataset, "ship" and "cat" should be quite different, but discriminator trained using MMD may overlook such differences (see Figure 4).
>
> Q4: The Literature review on other loss functions does not belong to the main body of the text.
> A4: We moved the literature review to Appendix B.1.
>
> Q5: What does it mean by assuming linear activation is used at the last layer of D.
> A5: We mean there is no activation function applied to the discriminator outputs. In the case of minimax and non-saturating loss functions, we could absorb the sigmoid function into the loss which results in the formation of softmax function.
>
> Q6: No need to include Arjovsky et al. (2017)'s statement on perfect discriminator.
> A6: We agree with the reviewer and deleted the statement.
>
> Q7: Why propose a generalized power iteration method when singular values can be computed as in [3]?
> A7: When only the first singular value is needed, the power iteration used in our study and [3] is computationally simpler than the method in [4] which uses Fourier transform and SVD. However, the strength of [4] is that all singular values can be computed in a single run, which may eventually inspire more powerful regularization methods for GAN. We discussed this in Appendix C.1 in the revised manuscript.
>
> Q8: “MS-SSIM is not compatible with CIFAR-10 and STL-10 which have data from many classes”; just calculate Intra-class MS-SSIM for CIFAR-10 and STL-10.
> A8: We deleted the statement in the revised manuscript.
>
> Q9: Should FID be used to evaluate a model trained with an MMD-loss when the discriminator uses almost the same architecture as the Inception model in FID?
> A9: We would like to point out that all loss functions in our study were paired with plain DCGAN architecture (see Appendix Table S1 and S2), which is much simpler than the Inception model.
>
> Q10: Which models in Table 1 used the spectral norm?
> A10: Spectral normalization was applied for all models in Table 1. We highlighted this in Table 1 note 1 and Sec. 5.2 in the revised manuscript.
>
> -------------------------------------------------
> [1]: On Gradient Regularizers for MMD GANs. NIPS, 2018.
> [2] Are GANs Created Equal? A Large-Scale Study. NIPS, 2018.
> [3] Spectral Normalization for Generative Adversarial Networks. ICLR, 2018
> [4] The Singular Values of Convolutional Layers. Under review at ICLR 2019.

---

### Author Response · Authors · 2018-10-04
**Code for MMD-GAN with the repulsive loss function**

Dear readers,
The code (and some raw results) for this paper can be found at the anonymized GitHub repository:
https://anonymous.4open.science/repository/e8675209-4393-4dbc-ad04-aad36cd5d738/
Any suggestion/feedback on the paper and code is much appreciated.

---

### Public Comment · (anonymous) · 2018-10-06
**questions**

Dear authors,

- Do similar results with Appendix A hold when we use the proposed repulsive loss?
- The calculation of the spectral norm does not appear novel to me. [1] offers an efficient and exact calculation of the spectral norm, and [2, 3] proposed the generalized power iteration. But my concern is rather a computational cost than the novelty. Appendix B reports no performance improvements over the original method [4]. Concerning computational cost and memory consumption, the original method is superior.  Are there any reasons to estimate the true spectral norm with paying additional overheads?

Thanks,

[1] Hanie Sedghi, Vineet Gupta, Philip M. Long.  The Singular Values of Convolutional Layers. arXiv 1805.10408, 2018
[2] Yusuke Tsuzuku, Issei Sato, Masashi Sugiyama. Lipschitz-Margin Training: Scalable Certification of Perturbation Invariance for Deep Neural Networks. NIPS, 2018
[3] Kevin Scaman, Aladin Virmaux. Lipschitz regularity of deep neural networks: analysis and efficient estimation. NIPS, 2018
[4] Takeru Miyato, Toshiki Kataoka, Masanori Koyama, Yuichi Yoshida. Spectral Normalization for Generative Adversarial Networks. ICLR, 2018

---

> ### Author Response · Authors · 2018-11-26
> **Local stability of repulsive loss and rational for estimation of real spectral norm**
>
> Thank you very much for your valuable comments. We try to address your concerns as below.
>
> Q1: Appendix A demonstrates the local stability of MMD loss. Are the results applicable to the proposed repulsive loss?
> A1: Yes.
> For the realizable case (the equilibrium P_X = P_G), we explicitly state the local stability of MMD-GAN using the repulsive loss in Appendix A.1 and proved it in Appendix A.3.
> For the non-realizable case (the real sample distribution is impossible to be fit by the generator), we used a simulation study (Figure S1) to show that both MMD loss and repulsive loss may be locally exponentially stable near equilibrium.
>
> Q2: What is the point of estimating the true spectral norm given the additional computational cost and no improvements?
> A2: Since the first submission, we've added experiments to compare the proposed power iteration for convolution (PICO) against the one on a matrix (PIM) [4]. The results can be found in Appendix C. In summary,
> 1) the spectral norm estimated by PIM may vary in a range related to the spectral norm by PICO;
> 2) PIM impose an indefinite and often loose upper bound on the Lipschitz constant of discriminator;
> 3) PICO performed better than PIM on cases using repulsive loss.
>
> Compared to PIM, PICO has a higher computational cost, which roughly equals the additional cost incurred by increasing the sample size by two. We recommend using PICO when the repulsive loss is used and the computational cost is less of a concern.
>
> Regarding the novelty, we notice that our proposed PICO is similar to that of [2][3], which we were not aware of during this study. We mentioned [1][2][3] as related work in the revised manuscript.
>
> ---------------------------------------------
> [1] Hanie Sedghi, Vineet Gupta, Philip M. Long.  The Singular Values of Convolutional Layers. arXiv 1805.10408, 2018
> [2] Yusuke Tsuzuku, Issei Sato, Masashi Sugiyama. Lipschitz-Margin Training: Scalable Certification of Perturbation Invariance for Deep Neural Networks. NIPS, 2018
> [3] Kevin Scaman, Aladin Virmaux. Lipschitz regularity of deep neural networks: analysis and efficient estimation. NIPS, 2018
> [4] Takeru Miyato, Toshiki Kataoka, Masanori Koyama, Yuichi Yoshida. Spectral Normalization for Generative Adversarial Networks. ICLR, 2018

---

### Author Response · Authors · 2018-11-30
**Summary of revision to the manuscript**

We thank the reviewers and area chair for their thoughtful comments and hard work, which we believe have contributed significantly to the improvement of our work. Here we summarize the changes to the manuscript:
1. In Appendix A, we added a proof of the local stability of MMD-GAN trained using the proposed loss, as requested by a public reader.
2. In Appendix C, we added a detailed comparison of the proposed power iteration method for convolution kernel against the one used in [1], as requested by Reviewer 2 and the public reader.
3. In Appendix D.1, we added an experiment exploring the repulsive loss with gradient penalty, as requested by Reviewer 1 and 3.
4. In Appendix D.2, we added an experiment exploring the effects of discriminator output dimension on the performance of proposed loss, as requested by Reviewer 2.
5. We revised the text to clarify our ideas and highlight important information in the experiment design and results.

For more information, please read our answers to each individual reviewer thread below.

We would also like to mention that the code (and some raw results) for this work can be found at the anonymized GitHub repository:
https://anonymous.4open.science/repository/e8675209-4393-4dbc-ad04-aad36cd5d738/

Thank you very much for reading. Any feedback on the manuscript and code will be much appreciated.

-------------------------------------------------
[1] Spectral Normalization for Generative Adversarial Networks. ICLR, 2018

---

### Meta-Review · Area_Chair1 · 2018-12-14

**Confidence:** 4
**Recommendation:** Accept (Poster)

**Metareview:**

The submission proposes two new things: a repulsive loss for MMD loss optimization and a bounded RBF kernel that stabilizes training of MMD-GAN. The submission has a number of unsupervised image modeling experiments on standard benchmarks and shows reasonable performance. All in all, this is an interesting piece of work that has a number of interesting ideas (e.g. the PICO method, which is useful to know). I agree with R2 that the RBF kernel seems somewhat hacky in its introduction, despite working well in practice.

That being said, the repulsive loss seems like something the research community would benefit from finding out more about, and I think the experiments and discussion are sufficiently extensive to warrant publication.